

# 1  Assessing the perturbations of the hydrogeological regime in

# 2  sloping fens through roads

Fabien Cochand[1], Daniel Käser[1], Philippe Grosvernier[2], Daniel Hunkeler[1], Philip Brunner[1]
[1]Centre of Hydrogeology and Geothermics, Université de Neuchâtel, Switzerland.
[2]LIN'eco, ecological engineering, PO Box 51, 2732 Reconvilier, Switzerland.
Corresponding author: Fabien Cochand, fabien.cochand@unine.ch
**Abstract**
Roads in sloping fens constitute a hydraulic barrier for surface and subsurface flow. This can lead to a
drying out of downslope areas of the sloping fen as well as gully erosion. Different types of road construction have
been proposed to limit the negative implications of the roads on flow dynamics. However, so far no systematic
analysis of their effectiveness has been carried out. This study presents an assessment of the hydrogeological
impact of three types of road structures in semi-alpine, sloping fens in Switzerland. Our analysis is based on a
combination of field measurements and fully integrated, physically based modelling. In the field approach, the
influence of the road was examined through tracer tests where the upslope of the road was sprinkled with a saline
solution. The spatial distribution of electrical conductivity downslope provided a qualitative assessment of the
flow paths and thus the implications of the road structures on subsurface flow. A quantitative albeit not site-specific
assessment was carried out using numerical models simulating surface and subsurface flow in a fully coupled way.
The different road types were implemented in the model and flow dynamics were simulated for a wide range of
slopes and hydrogeological conditions such as different hydraulic conductivity of the soil. The results of the field
and modelling analysis are coherent. Roads designed with an L-drain collecting water upslope and releasing it in
a concentrated manner downslope constitute the largest perturbations. The other investigated road structures were
found to have less impact. The developed methodologies and results are useful for the planning of future road
projects.





## 1 Introduction


Wetlands can play a significant role in flood control (Baker, 2009;Zollner, 2003;Reckendorfer, 2013),
mitigate climate change impacts (Cognard Plancq et al., 2004;Samaritani et al., 2011;Lindsay, 2010;Limpens,
2008) and feature great biodiversity (Rydin, 2005). However, the world has lost 64% of its wetland areas since
1900 and an even greater loss has been observed in Switzerland (Broggi, 1990). Therefore, wetland conservation
has received considerable attention. However, the sprawl of human infrastructure, land use changes, climate
change or river regulations remain serious factors that threaten wetlands. For instance, roads can substantially
modify the surface-subsurface flow patterns of sloping fens. The changes in flow patterns can influence sediment
transport, moisture dynamics and biogeochemical processes as well as ecological dynamics.
The link between hydrological changes and sediment dynamics has been studied in various contexts. From
a civil engineering perspective, erosion of the road must be avoided. A common strategy to avoid erosion of the
road foundation is to collect water in drains and then release it in a concentrated manner downslope. This, however,
can lead to erosion of the downslope area, a phenomenon known as « gully erosion ». A number of studies
specifically focused on identifying the controlling processes and relevant parameters of gully erosion (Capra et al.
(2009);Valentin et al. (2005);Descroix et al. (2008);Poesen et al. (2003);Martínez-Casasnovas (2003);Daba et al.
(2003);Betts and DeRose (1999);Derose et al. (1998), among others). Nyssen et al. (2002) investigated the impact
of road construction on gully erosion in the northern Ethiopian highlands, with a focus on surface water. In their
study area, they observed the formation of a gully after the road construction downslope culvert and outlets of
lateral road drains. Based on fieldwork and a subsequent statistical analysis, they concluded that the main causes
for gully development are the concentrated runoff, the diversion of concentered runoff to other catchments and the
modifications of drainage areas induced by the road. The role of groundwater was not considered in this study.
Reid and Dunne (1984) developed an empirical model for estimating road sediment erosion of roads located
in forested catchments in the Washington state (USA). They concluded that a heavily used road produced 130
times more sediment that an abandoned road. Wemple and Jones (2003) also developed an empirical model for
estimating runoff production of a forest road at a catchment scale. They demonstrated that during large storm
events, subsurface flow can be intercepted by the road. The intercepted water, if directly routed to ditches, increases
the rising limb of the catchment hydrograph. At a smaller spatial scale (0.1 km$^2$) Loague and VanderKwaak (2002)
assessed the impact of a road on the surface and subsurface flow using an integrated surface-subsurface flow model
InHM (Integrated Hydrology Model) (VanderKwaak, 1999) in a rural catchment. The results showed that the road





induced a slight increase of runoff and a decrease of surface-subsurface water exchange around the road. Dutton
et al. (2005) investigated the impact of roads on the near-surface subsurface flow using a variability saturated
subsurface model. They concluded that the permeability contrast caused by the road construction leads to a
disturbance of near-surface subsurface flow which may significantly modify the physical and ecological
environment.
Road construction can also impact the development of vegetation (Chimner, 2016). von Sengbusch (2015)
investigated the changes in growth of bog pines located in a mountain mire in the black forest (south-west
Germany). The author suggests that the increase of bog pine cover is caused by a delayed effect of a road
construction in 1983 along a margin of the bog. The road affects subsurface flow and therefore prevents the upslope
water to flow to the bog. According to von Sengbusch (2015), the road disturbances induce a larger variability in
water table elevations during dry periods and consequently increase the sensitivity of the bog to climate change.
Based on these previous studies and basic principles of subsurface flow, a simple conceptual model
describing the influence of roads on the flow system can be drawn (Figure 1). Roads are generally built with
materials of low hydraulic conductive and therefore act as a hydrogeological barrier. In natural conditions,
rainwater infiltrates the soil and follows the topographical gradient. In case of heavy precipitation events, water
can also directly flow on the surface (Figure 1a). If a road is constructed, it constitutes a hydrogeological barrier
(Figure 1b) and consequently affects the flow dynamics. Drains installed underneath the road Figure 1c) can
mitigate the effect of this hydrogeological barrier. The design and the materials of drains significantly affect flow
dynamics. Figure 1c presents a typical condition where a non-continuous drain (i.e., drains are perpendicularly
installed at regular distances along the road) is used to connect both sides of the road. Upstream and downstream
subsurface flows are deviated and the drain becomes the main outlet. The concentration of subsurface flow
downstream of the drain may induce gully erosion and disturb the hydraulic regime of the sloping fens.





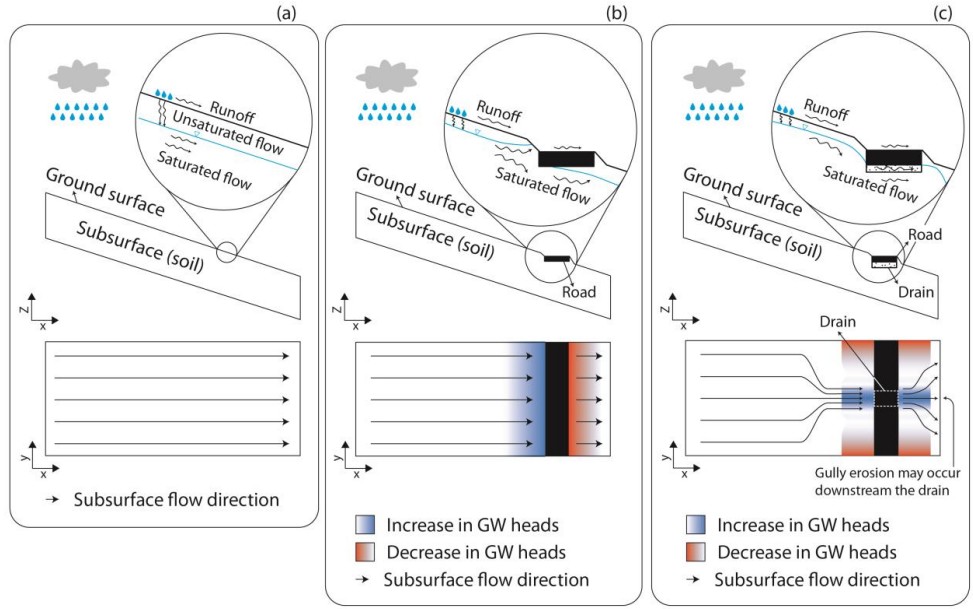

**Figure 1 Conceptual subsurface dynamics in sloping fens: a) natural conditions, b) with road without a drain and c)**
**with a road and a drain.**
While these studies clearly indicate that roads can have adverse effects on the surface and subsurface flow
dynamics and the associated ecosystems, a detailed study on how roads perturb the flow system and dynamics in
a sloping fen has not been carried out. In Switzerland, more than 20'000 ha are included in the national inventory
of fens of national importance (Broggi 1990), most of them are located in the mountainous regions of the northern
Prealps. Hence, the majority of Swiss fens is composed of sloping fens, which developed on nearly impermeable
geomorphological layers such as silty moraine material or a particular rock layer named "flysch". Although
organic, soils are not necessarily peaty and most of the time quite superficial, not exceeding a few decimeters in
thickness. Water flow is therefore mostly consisting of runoff and partly occurring in the shallow part of the
subsurface. The construction of a road in this kind of sloping fens removes completely the soil layer in which
subsurface flow occurs, thus constituting a major perturbation of the hydraulic regime. Construction techniques to
limit these adverse impacts have been proposed but their efficiency has so far not been investigated. Three road
structures with various construction techniques and materials (hereinafter further detailed) were developed in
Switzerland to reduce impacts of roads. These road types are conceptually illustrated in Figure 2. The efficiency
of developed road structures was so far not assessed after completion. This study focuses on these three road
structures described hereafter:





- The *no-excavation* structure (Figure 2a) aims at preserving soil continuity under the road. It consists of a levelled layer of gravel, anchored to the ground, and underlying 0.16 m thick concrete slabs. Soil compaction is limited by using a low-density gravel, made of expanded glass chunks (Misapor™) - approximately fivefold lighter than conventional material.

- The *L-drain* structure (Figure 2b) aims at collecting subsurface water upstream the road and redirecting it to discrete outlets on the other side. The setup consists of a trench, approximately 0.4 m deep, filled with a matrix of sandy gravel that contains an L-shaped band of coarse gravel acting as the drain.

- The *wood-log* structure (Figure 2c) aims at promoting homogeneous flow under the road but does not preserve soil continuity. Embedded in a trench, approximately 0.4 m deep, the wooden framework is filled with wooden logs forming a permeable medium. The wooden logs are then covered with mixed gravel.

The aim of this study is to investigate, document and assess the hydrogeological impact of various road structures and their effects on fen water dynamics. A combination of fieldwork and hydrogeological modelling tasks was employed. Fieldwork was used to document and obtain the required information on the hydrogeological impact of existing road structures on fen water dynamics. Sites with similar natural conditions were chosen to compare the influence of different road constructions on flow processes. The field studies allow for assessing the effectiveness of a given road structure, however, they cannot provide generalizable analysis of the different road types under different environmental and physical conditions, e.g. the slope or the hydraulic properties of the fen. This gap was filled by the development of generic numerical models. The main advantage of the modelling approach is the possibility to generate a multitude of different models with various characteristics such as different road structures, slopes or fen hydraulic conductivity and to test their impacts on the flow dynamics.



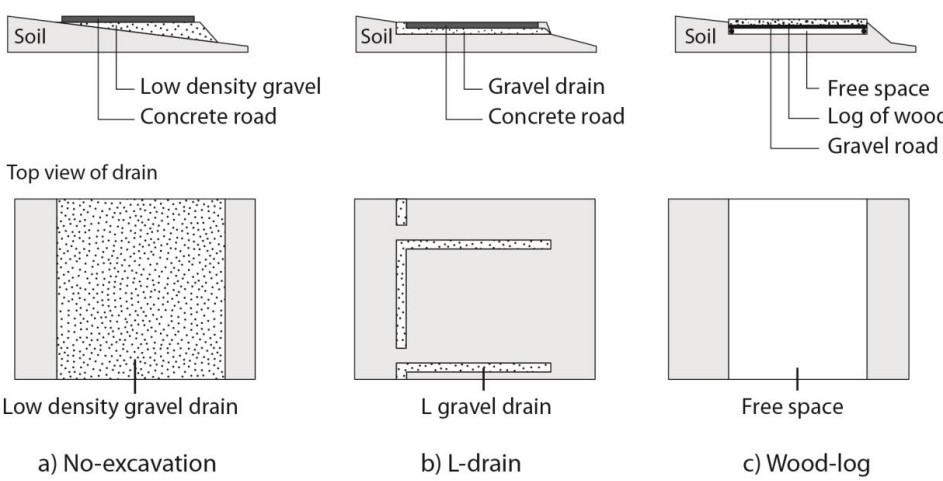

**Figure 2 : Conceptual road structures, a) No-excavation road structure, b) L-drain road structure and c) Wood-log road structure.**

## 2    Methods

### 2.1    Study areas and fieldwork

Four sloping fen areas located in alpine or peri-alpine regions of Switzerland (Table 1) were selected. All areas are situated in protected fen areas, and their selection was based on two main criteria:

1.  The subsurface water flow must occur only in the topsoil layer and as runoff (as described in the introduction).

2.  The types of installed road structures (no-excavation, L-drain and wood-log).

To fulfil the first criteria, soil profiles were analysed to ensure that each area with different road types had the same soil stratigraphy: It had to be composed of organic soil on top of a layer of impermeable clay and similar hydraulic regimes (e.g., runoff and subsurface flow occurring only in the topsoil layer). In addition, to ensure that subsurface water is forced to cross the road instead of flowing in parallel of the road (and thus not being affected directly by the road), another important criterion for the selection of the study areas was that subsurface flows perpendicular to the road.

To evaluate the hydraulic connection provided by the road structures, tracer tests were carried out. As illustrated schematically in Figure 3, a saline solution was spread on the upslope area and the occurrence of the



tracer was monitored downslope the road. In the absence of surface runoff, the occurrence of a tracer downslope
demonstrates the hydrogeological connection through the road. Furthermore, the spatial distribution of the tracer
front reflects the heterogeneity of the flow paths.
**Table 1. Field site locations and features.**

|  | St-Antonien (STA) | Schoeniseischwand (SCH) | Stouffe (STO) | Höhmad (HMD) |
|---|---|---|---|---|
| *Road type* | No excavation | L-Drain | Wood-log | Wood-log |
| *Terrain slope* | 0.27 | 0.13 | 0.13 | 0.15 |
| *WGS84 coordinates* | 46.96760°N 9.84843°E | 46.78872°N 7.96805°E | 46.72957°N 7.83861°E | 46.74027°N 7.89871°E |


Each area corresponds to an 8 x 20 m rectangle that includes a 2.5 to 3.5 m wide road segment. A network
of approximately 30 mini-piezometers on both sides of the road (Figure 3) was installed to monitor the hydraulic
heads and was used to obtain samples for the tracer test.
The mini-piezometers are high-density polyethene (HDPE) tubes no longer than 1.5 m (ID: 24 mm). Each
tube was screened with 0.4 mm slots from the bottom end to 5 cm below ground level. It was inserted into the soil
after extracting a core with a manual auger (diameter: 4-6 cm). The gap between the tube and the soil was filled
with fine gravel and sealed on the top with a 4 cm thick layer of bentonite or local clay. Hydraulic heads were
measured using a manual water-level meter (± 0.3 cm). At each point, the terrain and the top of the piezometer
were levelled using a level (± 0.3 cm), whereas the horizontal position was measured with a tape measure (± 5
cm).
The tracer tests were conducted using two oscillating sprinklers designed to reproduce a 30 mm rain event
during 2-3 hours. This is equivalent to an intense rain event. Prior to the experiment, the sprinklers were activated
for 15-60 minutes to wet the soil surface. Sodium chloride was added to the irrigated solution to obtain an electrical
conductivity of 5-10 mS/cm which is approximately ten times higher than the natural electrical conductivity of the
groundwater. Then, the area (60 m$^2$) upslope of the road (upslope injection area of Figure 3) was irrigated with the
salt solution using the two sprinklers. The electrical conductivity (EC) of soil water was manually measured using
a conductimeter in all mini-piezometers prior to the experiment, immediately after, and 24h later. An increase in
EC in piezometers located in the downslope area indicates that the injected salt water flowed from the upslope
area to the downslope area below the road and indirectly indicates a hydraulic connection. Conversely, if no




changes in EC are observed in piezometers, this indicates that there is no connection below the road and finally a
decrease in EC is not expected.

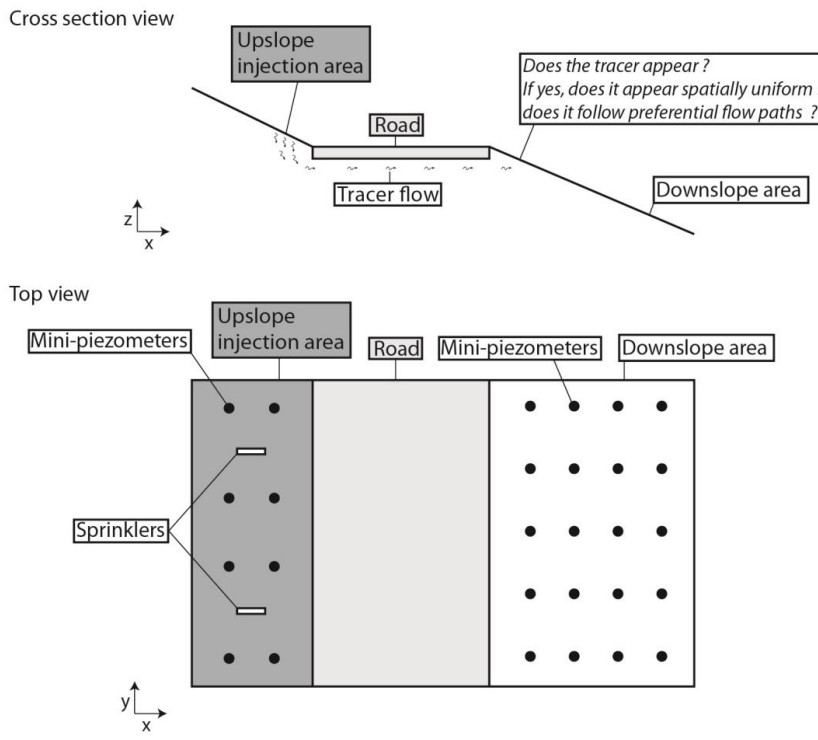


**Figure 3 : Schematic view of the fieldwork areas.**
**2.2    Numerical modelling**

162        To simulate and quantify the impact of the roads on the flow dynamics in sloping fens, the modelling

approach was structured in three steps. First, a 3D base case model representing surface and variably saturated
subsurface water flow in a sloping fen was elaborated. Then, the base case model was modified to represent the
three different types of investigated road structures. For each model, various slopes, organic soil and road drain
hydraulic conductivities were implemented to produce a sensitivity analysis and explore their sensitivities in the
sloping fen flow dynamics (see section 2.2.3 for details). Finally, a comparison of all model results was made in
order to assess the impact of road structures and quantify the dynamics and the physical controls of subsurface
flow in these environments.



**2.2.1 Numerical simulator**

171        The model used in the study is HydroGeoSphere (HGS) (Therrien et al., 2005). HGS is a physically-based

surface–subsurface fully-integrated model using the control volume finite element approach. HGS simultaneously
solves a modified Richards' equation (Eq. 1) describing the saturated and unsaturated 3D subsurface flow and the
2D depth average diffusion-wave approximation of the Saint Venant equation (Eq. 3) for describing the surface
flow. Richard's equation is given as:

$$-\nabla \cdot (q) \pm Q_s + \sum \Gamma_{ex} = \frac{\partial}{\partial t}(\theta_s S_w)$$

**Eq. 1**

where $\nabla = \partial/\partial x, \partial/\partial y, \partial/\partial z$, $Q_s$ represents fluid exchanges with the outside of the simulation domain (*e.g.*
injection wells), $\sum \Gamma_{ex}$ is the volumetric fluid exchange rate between the subsurface domain and all other simulation
domains supported by HGS (surface domain, tile drain domain, among others), $\theta_s$ is the porosity, $S_w$ is the water
saturation and q is the Darcy flux (Eq. 2) of water given as:

$$q = K \cdot k_r \nabla(\psi + z)$$

**Eq. 2**

Where K is the hydraulic conductivity of the subsurface $\psi$ is the pressure head of water, z is the elevation and $k_r$
is the relative hydraulic conductivity. $k_r$ varies between 1 when the domain is fully saturated and near zero when
the domain is fully unsaturated. The Van Genuchten (1980) functions which relate pressure head to saturation and
relative hydraulic conductivity is employed. For surface flow, the diffusion-wave approximation of the Saint
Venant equation (Eq. 3) is given as:

$$\frac{\partial \phi_o h_o}{\partial t} - \frac{\partial}{\partial x}\left(d_o K_{ox} \frac{\partial h_o}{\partial x}\right) - \frac{\partial}{\partial y}\left(d_o K_{oy} \frac{\partial h_o}{\partial y}\right) + d_o \Gamma_o \pm Q_o = 0$$

**Eq. 3**

where $\phi_o$ is the surface porosity varying between zero at the ground surface and unity at the top of a rill or
obstruction, $h_o$ is the surface water elevation, $d_o$ surface water depth ($h_o = d_o + z$), $Q_o$ is a volumetric flow rates
representing external source and sinks, $K_{ox}$ and $K_{oy}$ are surface conductance (Eq. 4) given as:



$$K_{ox,y} = \frac{d_o^{2/3}}{n_{x,y}} \frac{1}{(\partial h_o / \partial s)^{1/2}}$$

**Eq. 4**

where $n_{x,y}$ are the Manning roughness coefficients in the x and y directions and s is the slope taken in the direction
of maximum slope. Finally, the term $\Gamma_o$ represents fluid exchanges with other domains. The water exchanges
between the surface and subsurface domains are calculated using the "dual node approach" (Eq. 5) and is given
as:

$$d_o \Gamma_o = \frac{k_r K_{zz}}{l_c} (h - h_o)$$

**Eq. 5**

In this approach, the top nodes representing the ground surface are used for calculating both subsurface and surface
flow. The water exchanges are calculated as hydraulic head differences of the two domains represented by $h - h_o$
and multiplied by the vertical hydraulic conductivity of the top layer, $k_r K_{zz}$, and a coupling factor, $(l_c)$.
The iterative Newton-Raphson method is used to solve the nonlinear equations. At each subsurface node, saturation
and groundwater heads are calculated, which allows for the calculation of the Darcy flux. On the surface domain,
the surface water heights are calculated at each node to determine surface water flux. Rivers and lakes are
characterized by a surface water depth larger than 0. For further details on the code and HGS capabilities, see
Therrien et al. (2005), Li et al. (2008) or Brunner and Simmons (2012).
**2.2.2    Conceptual models and model implementation**

201         Figure 4 illustrates the conceptual model of each case. Geometry, topography and slopes are based on the

physical conditions in the field. In each model, the soil layer has a thickness of 0.4 m and the surface and subsurface
water are only supplied by precipitation. The upstream boundary is the catchment boundary (water divide) and the
downstream boundary represents the outlet of the model. Finally, it was assumed that the layer beneath the soil
was impermeable (as observed in the field) and engineering plans were used to design drain and road. One
Neumann (constant flux) boundary condition was used on the top face for simulating precipitation. A constant
groundwater head boundary condition (Dirichlet type) equal to the ground surface elevation (2m) was used on the
right face (x=76 m on the Figure 5a) allowing the groundwater to flow out of the model. Finally, a critical depth
boundary condition which forces the surface water to reach a given elevation (2 meters in our case) to flow out of
the model was implemented on the top nodes located at x=76 m and all other faces are no flow boundary conditions.



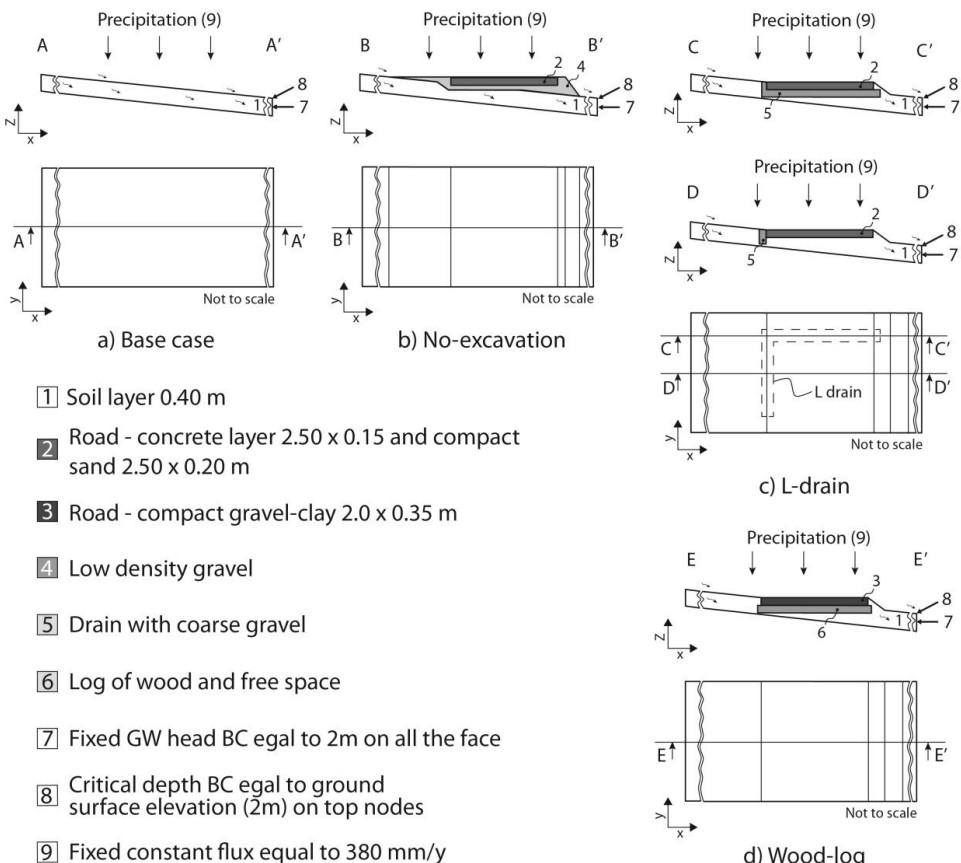


**Figure 4 : a) Base case, b) No-excavation, c) L-drain and d) Wood-log structures conceptual models.**
To numerically solve the 3D flow equation, a 3D mesh was developed (Figure 5a). The mesh is 76 m long
in the X direction, 20 m in the Y direction and the mesh thickness is 1.2 m. The top elevation was fixed at 2 m on
the right side (x=76m) and varies from 9.6 m to 24.8 m on the left side (x=0) according to the slope of the model.
The mesh was made up of 24 layers, 127,200 nodes and 118,440 rectangular prism elements. To ensure an
appropriate level of detail, several mesh discretization refinements were made. Therefore, the element size varies
between 2m and 0.1 m horizontally (in the X and Y directions) and 0.09 m and 0.05 m vertically.
The base case model and the three other models representing different road types have the same boundary
conditions and finite element meshes, however, modifications were made between coordinates 61<x<66 to
implement the different road types. Figure 5 depicts the differences between the base case model (Figure 5a and
b) and models with roads (Figure 5c, d, e and f). In the case of models with a road, the mesh was deformed and


the properties were changed. The fine spatial discretization of the mesh created between the coordinates 61<x<66
allows a more accurate representation of the simulated processes where high hydraulic gradients are expected (near
roads and drains). Additionally, the refinements allow an accurate representation of drains and the roads.

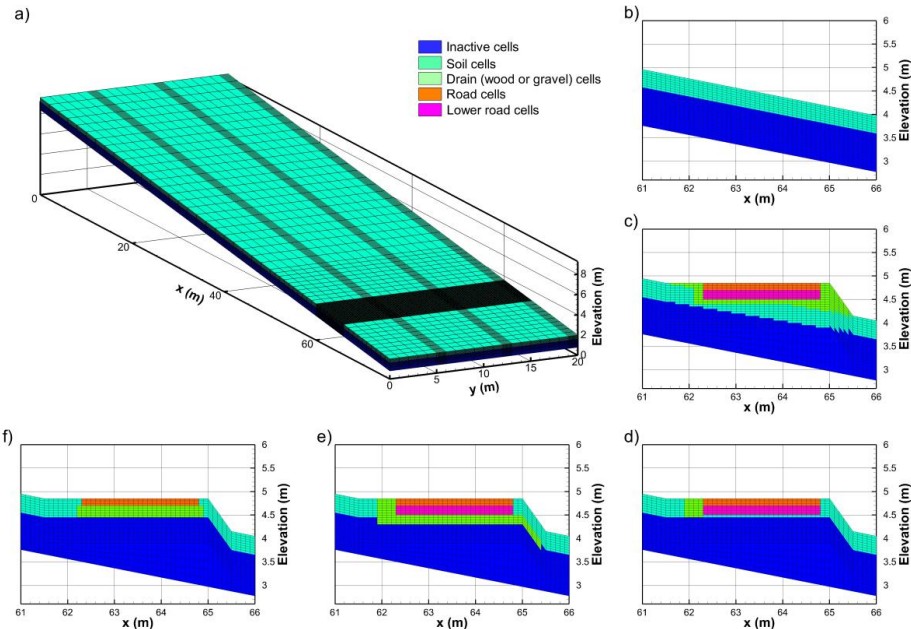


**Figure 5 : Model development: a) Base case model, b) Base case model cross-section between 61m < x < 66m, c) No-**
**excavation model between 61m < x < 66m, d) L-drain model between 61m < x < 66m, e) L-drain model between 61m <**
**x < 66m along the transversal drain f) Wood-log model between 61m < x < 66m.**
**2.2.3    Sensitivity analysis**

231        The sensitivity analysis consists of the variation of model properties and parameters in order to understand

how they control the sloping fen dynamics. The sensitivities of the following parameters were analyzed: fen slope,
soil hydraulic conductivities and road drain hydraulic conductivities. For each property, three different values were
selected and are summarized in Table 2. The soil hydraulic conductivities vary between 8.64 [m/d] and 0.0864
[m/d] to represent all ranges of observed soil hydraulic conductivities (Charman, 2002). The road drains hydraulic
conductivities vary between 8640 [m/d] and 86.4 [m/d] which correspond to very coarse and coarse gravel. Finally,
the slopes were fixed at 10%, 20% and 30% as observed during the fieldwork. Note that the drain hydraulic
conductivities of the wood-log (W-L) were assumed ten times more conductive and more porous than gravel drain
because of its particular structure (wood logs). In order to simulate each parameter combination, a total of 90
models were developed (27 models for each road structures and 9 models for natural conditions).





**Table 2 : Subsurface and surface flow parameters.**

| Subsurface flow properties | | | | | |
|---|---|---|---|---|---|
| | **Hydraulic conductivity** | **Porosity** | **Van Genuchten α** | **Van Genuchten β** | **Residual water content** |
| **Units** | K [md⁻¹] | θ [-] | α [m⁻¹] | β [-] | Swr [-] |
| **Soil - KS1** | 8.64 | 0.25 | 4 | 1.41 | 0.04 |
| **Soil - KS2** | 0.864 | 0.25 | 4 | 1.41 | 0.04 |
| **Soil - KS3** | 0.0864 | 0.25 | 4 | 1.41 | 0.04 |
| **Drains - KD1** | 8640 | 0.25 | 29.4 | 3.281 | 0.04 |
| **Drains - KD2** | 864 | 0.25 | 29.4 | 3.281 | 0.04 |
| **Drains - KD3** | 86.4 | 0.25 | 29.4 | 3.281 | 0.04 |
| **Drains - WL - KD1** | 86400 | 0.7 | 29.4 | 3.281 | 0.04 |
| **Drains - WL - KD2** | 8640 | 0.7 | 29.4 | 3.281 | 0.04 |
| **Drains - WL - KD3** | 864 | 0.7 | 29.4 | 3.281 | 0.04 |
| **Road concrete** | 0.0000864 | 0.05 | 1.581 | 1.416 | 0.04 |
| **Road fine sand** | 0.00864 | 0.25 | 4 | 1.416 | 0.04 |
| Surface flow properties | | | | | |
| | **Coupling length** | **Manning's roughness coefficient** | | **Rill storage height** | **Obstruction height** |
| **Units** | $l_c$ [m] | $n_x$ [m⁻¹ᐟ³s] | $n_y$ [m⁻¹ᐟ³s] | $D_t$ [m] | $O_t$ [m] |
| **Soil** | 1. x 10⁻² | 0.03 | 0.03 | 0.005 | 0.005 |
| **Road** | 1. x 10⁻² | 0.018 | 0.018 | 0.001 | 0.001 |


Models are run for 10'000 days (about 27 years) with a constant flux equal to 380 mm/y on the top
representing the rainfall to reach a steady state. This precipitation allows for the saturation of the downslope part
of the model. Subsequently, subsurface water velocities in the soil layer were extracted at each observation point
located downslope from the road (Figure 6). The subsurface water velocities are proportional to the volumetric
groundwater flow and consequently, changes in groundwater velocity indicate a perturbation of flow dynamics.
Therefore, a comparison of velocities between each model was made to present the effect of each road structure
and sloping fen properties on the dynamics.



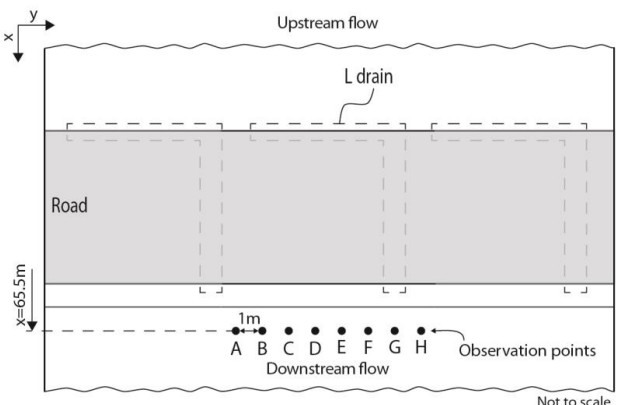


**Figure 6 : Location of observation points in the models.**

**3     Results and Discussion**

**3.1     Fieldwork**

Based on the observations, all sites show a continuous saturated zone before the experiment, both upstream and

downstream of the road, the hydraulic gradients being mostly similar to the terrain slope (Figure 7, 1st column).

In contrast, the EC maps established prior to the tracer test show spatial variability at a length scale of one to

several meters (Figure 7, 2nd column.). Within each plot, EC varies from 482 to 629uS/am. At the SCH site, the

highest values are located downstream of the L-drain outlet which could indicate that the EC increases as water is

flowing through the drain (e.g. through the dissolution of the construction material). Given that this initial

distribution of EC is not uniform, the comparison of EC after the sprinkling experiment has to be made in a relative

manner (Figure 7, 3rd column).

The heterogeneity of the hydraulic conductivity of the soil is apparent from the tracer test results (Figure 7,

3rd column: EC 24 hours after injection). At all four sites, the front of the saline solution is not uniform but follows

the heterogeneity of the soil hydraulic conductivity. Nevertheless, road structures may play the role of a

preferential flow path that is particularly obvious at the SCH site where the front follows two preferential flow

paths. One related to the L-drain (right path) and the other on the left, unrelated to the L-drain, suggesting that the

latter drains only a part of the water and the other part follows a natural preferential flow path. At the HMD site,

the saline solution is far more concentrated on the left side of the plot, yet apparently not as a result of the road's

structure. Rather, the soil appears more permeable on the left side of the plot, both upslope and downslope of the

road. Finally, the decrease in EC observed 24 hours after injection at some locations might result from the





following: (1) the tracer injection induces, by "piston effect", the displacement of a small volume of local water
with a lower EC; (2) the tracer injection was preceded by a period of irrigation without tracer, which could have
diluted the pre-irrigation soil solution.
In each case, the irrigation experiments demonstrate the continuity of subsurface flow under the road for
all structures. For the no-excavation and wood-log type, the perturbation of the flow field seems controlled by the
natural heterogeneity of the soil and flow paths, and not by the road itself. Conversely, the field data strongly
suggest that the L-drain constitutes an important preferential pathway and consequently subsurface flow is
increasingly concentrated. In terms of wetland conservation, this flow convergence is a serious threat (gully
erosion, local drying up of the soil). Despite these strong indications, it is clear that with the field data alone no
conclusive analysis can be made as no data before the construction of the road are available. Fieldwork allows for
site-specific conclusions, but more general conclusions which are not specific to a site are impossible. Therefore,
numerical modelling was used to fill this gap.








**Figure 7 : Fieldwork results at the four field sites: 1st column) Measured groundwater heads before tracer test, 2nd column) Measured EC before tracer test and 3rd before and after tracer test differences in EC.**






### 3.2 Modelling

Figure 8 shows the results of the models with a slope of 10%, Figure 9 with a slope of 20% and Figure 10 with a slope of 30%. In each dot chart, the groundwater velocities (always in m/d) are plotted with crosses for the base case model, diamonds for the no-excavation type, squares for the L-drain type and circles for the wood-log type. In following paragraphs, the base case (natural conditions) results are presented and discussed, followed by the simulations of the road structures. Finally, all model results are discussed.

In the base case model, groundwater velocities vary from 0.013 (m/d) to 0.269 (m/d) for 10% slope, 0.025 (m/d) to 0.269 (m/d) for 20% slope and to 0.038 (m/d) to 0.274 (m/d) for 30% slope. The groundwater velocity decreases gradually depending on the hydraulic conductivities (KS) of the soil layer. For any slope, where hydraulic conductivities are high (KS1), groundwater velocities are higher compared to the case where hydraulic conductivities are low (KS3). The primary observation is that groundwater velocities are mainly controlled by the hydraulic conductivities and therefore the slope plays a minor role. Differences between the maximum and minimum hydraulic conductivity are two orders of magnitude, whereas changes between slopes are very small. The main effect of slope is to slightly increase groundwater velocities when the hydraulic conductivities are low (0.0864 m/d) due to the increase of the topographic gradient.

In the no-excavation and wood-log type models, the effect of road structures is quite similar. The groundwater velocities vary from 0.028 (m/d) to 0.269 (m/d) for 10% slope, 0.025 (m/d) to 0.269 (m/d) for 20% slope and 0.042 (m/d) to 0.269 (m/d) for 30% slope. Compared to the base case model, results show that the no-excavation and wood-log type structures have a minimal impact. The only marked difference is that groundwater velocities are slightly higher if the hydraulic conductivities are low (KS3) for each slope in the wood-log type model. This can, to a certain extent, be explained by the fact that the hydraulic conductivity of the base of the road (consisting of wood-logs) is higher than the hydraulic conductivity of the soil and therefore facilitate the infiltration. Conversely, in the base case model, less water is infiltrated but more runoff occurs. For the no-excavation model with a slope of 10%, results are not presented for technical reasons. For this specific geometry and topography, a different structure of the mesh had to be generated which did not allow for a direct visual comparison with the other models. In the 20% and 30% slope models, the results of the no-excavation model are similar to the base case model.

In the L-drain type model, the effect is markedly different from the other road structures. The groundwater velocities vary significantly in the observation points. The maximum velocities are always obtained in the





observation point G just downstream the drain outlet and may be 10 times higher than in the base case. Conversely,
minimum velocities are obtained in C and D observation points in which velocity may be 10 times lower.
Significant differences in groundwater velocity are also observed in the same transect. The maximum differences
are observed if the hydraulic conductivity of soil (KS) and drain (KD) are high and may vary from to 0.017 (m/d)
to 1.281 (m/d). Conversely, when KS and/or KD are low, the differences along the transect are smaller. The L-
drain structures also facilitate water infiltration in soil with a low permeability (KS3) where groundwater velocities
are slightly higher than the base case model. Finally, it can be seen that slope accentuates groundwater velocity
differences along the transect. In the scenario where the hydraulic conductivity of the soil and the drain are high
(KS1 and KD1), the differences in groundwater velocity for the 10% and 30% slope scenarios increase.

325        Results show that the no-excavation structure has the least impact on the groundwater velocities and the

wood-log structure has a limited impact on groundwater dynamics. The only difference with the base case (no road
at all) model is that the groundwater velocities observed are slightly higher where the hydraulic conductivity of
the soil layer is low (KS3). This is caused by the wood-log drain which facilitates water infiltration in a low-
conductive soil layer. Finally, the L-drain structure impacted significantly the groundwater dynamics. Significant
differences are observed in each scenario, mainly due to the L-shape drain. Downstream of the drain outlet
(observation point G), groundwater velocities are higher than other observation points along the transect,
regardless of the slope and the drain hydraulic conductivity. Maximum differences may reach two orders of
magnitude from 0.0346 (m/d) to 1.296 (m/d) in the same transect. Only the soil hydraulic conductivity reduces
differences in groundwater velocity along the transect and the slightly higher groundwater velocity in comparison
with the base case model indicates that gravel drain also facilitates water infiltration in low-conductivity soil layer.
The impact of the L-drain road structure which concentrates groundwater flow is clearly identified in the numerical
approach and is consistent with the field observations. For other road structures also, numerical models are
consistent with fieldwork results by showing a relatively undisturbed groundwater flow downslope the road. The
use of numerical models allowed for a quantitative estimation of the flow perturbation induced by each road
structure and model results were consistent with the field observations. In addition, the development of models
with various combinations of parameters also allowed for exploring a larger parameter space than using field work
only. For instance, the fact that the impact of an L-drain structure on the water dynamics is less marked if the
hydraulic conductivity of soil is low would have been impossible to identify by using fieldwork only.





The main simplification of the model is the assumption of a homogeneous hydraulic conductivity of the soil.
Groundwater flow in fens can occur along preferential pathways. Therefore, the models are not able to reproduce
small-scale processes, i.e. the exact hydraulic head in an individual mini-piezometer. Models results have to be
interpreted as an average across multiple preferential flow paths.
Further investigations should be carried out to identify groundwater velocity threshold values above which a risk
of for instance gully erosion is present. This is especially important for L-drain structures where the increase of
flow velocities is higher than for the other structures. Finally, the impact on sloping fen vegetation related to
perturbations of the groundwater flow should be further investigated. In this way, road construction could be better
planned.





**Figure 8 : Simulated groundwater velocities downslope each road structures and each parameter combination with a**
**slope of 10%.**

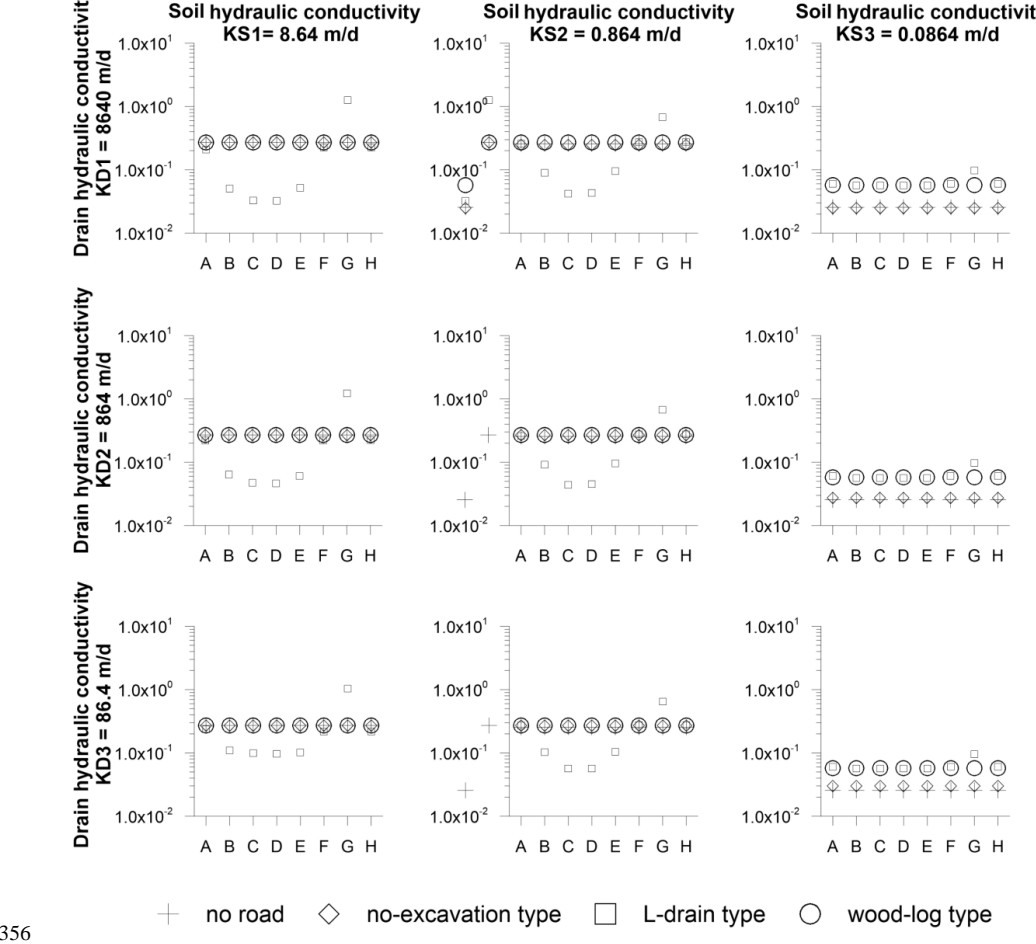


**Figure 9 : Simulated groundwater velocities downslope each road structures and each parameter combination with a**
**slope of 20%.**





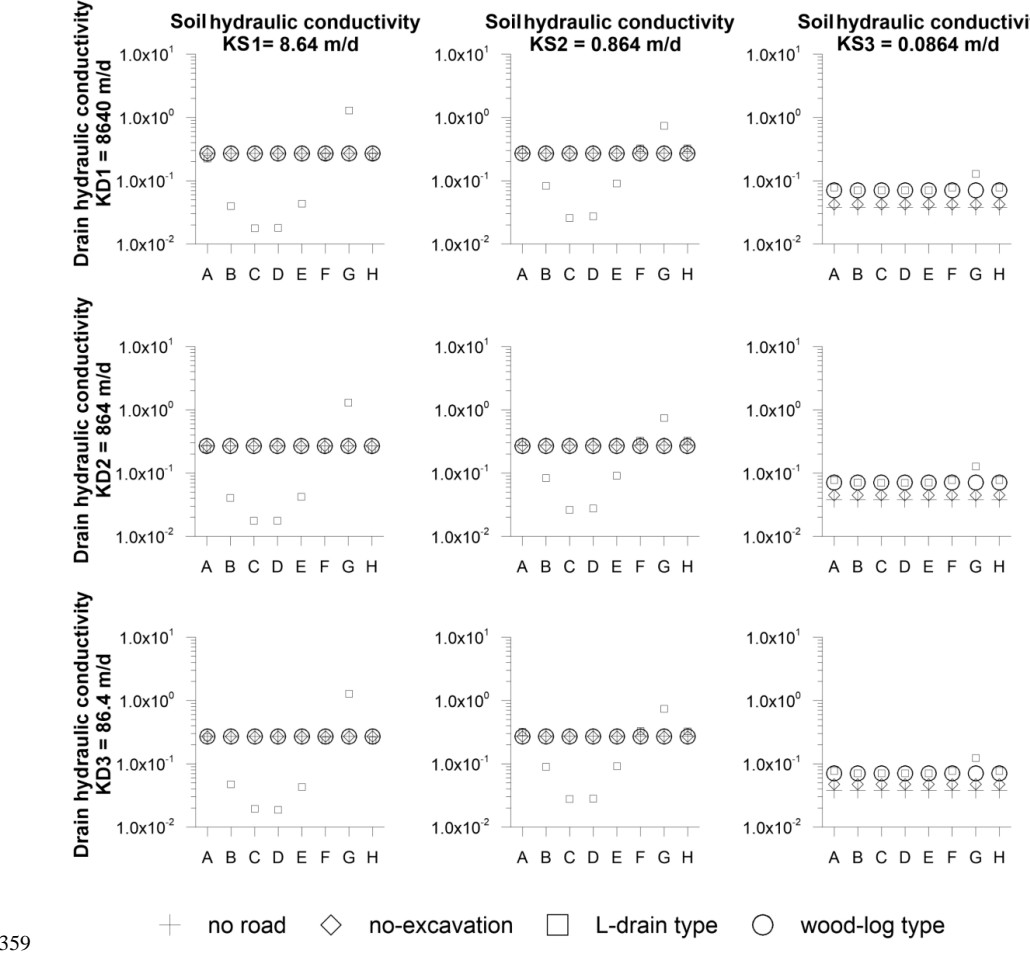

**Figure 10 : Simulated groundwater velocities downslope each road structures and each parameter combination with a slope of 30%.**

## 4    Conclusions

This study presented an assessment of three road structures regarding their perturbations of the natural

groundwater flow. Two of these road structures were specifically developed to reduce the negative impacts of the

road. The study is based on two complementary approaches; a tracer test in the field and numerical models

simulating groundwater flow for the different road structures.

The tracer tests showed that both sides of all investigated road structures were hydraulically connected.

Groundwater flow was heterogeneous suggesting the occurrence of preferential flow paths in the soil. The presence

of a transversal drain (L-drain) beneath the road constitutes a preferential flow path, however, which is of much



greater importance than the naturally occurring preferential pathways. This was also confirmed by the models.
Velocities 10 times larger than in the natural case were obtained in the numerical simulations. This is not further
astonishing as the drains were specifically designed for this purpose. The two other road structures (wood-log and
no-excavation) do not perturb the flow field to the extent of the L-drain. To minimize the perturbation of flow
fields, the wood-log and no-excavation structures are recommended.
The combination of fieldwork and the development of numerical models was fundamental to achieve the
goal of this study. The tracer test allowed for a better understanding of groundwater flow throughout road structures
and allowed for evaluating their effectiveness at a given location. However, the tracer tests are time-consuming
and only a few field sites are available. The numerical approach, on the other hand, allows for exploring any
combination of slope, hydraulic properties or road structure, thus providing a more comprehensive approach. In
our study, the trends between the numerical and field approaches were consistent.

## 381 5    Acknowledgements

This research was funded by the Swiss Federal Office for the Environment (FOEN) and supported by Swiss
Federal Office for Agriculture (FOAG). The authors are grateful to Benoit Magnin, Peter Staubli Andreas Stalder,
Anton Stübi and Ueli Salvisberger for their collaborations.

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
