# Peer review of "Assessing the perturbations of the hydrogeological regime in sloping fens through roads"

_Hydrology and Earth System Sciences, 2018_

## Referee Comment (RC1) · Anonymous Referee #1 · 12 Dec 2018

This manuscript presents an assessment study of the hydrogeological impact of road construction with different drainage designs applied to sloping fens, using combined field and numerical analyses. The study presented is comprehensive, thorough, well-organized, and clear. Among other things, this study presents one of the best examples on how we can combine field and numerical studies in hydrogeological analyses. The conclusions are informative and clear.

As it was stated in the manuscript, comprehensive field tests could be the best to better understand the problems in general. However, they are time-consuming and only a limited number of sites are available to accommodate the tests. The numerical analy-

sis, on the other hand, allows us to explore and assess multiple designs with levels of variations relatively easily, yet the validity needs to be carefully checked. The authors compare the field study and numerical results to ensure that the integrated hydrologic simulator is an appropriate and meaningful tool to reproduce the observations made through field tests, and also to evaluate the alternative designs. Based on the comprehensive evaluation of the three drainage designs, the authors concluded that the road construction with L-drain is the most vulnerable in terms of potential soil erosion underneath the roads. I do not see any over-statement in the conclusions drawn, and it seems to be a very useful guideline to be considered by practitioners.

———————————————

---

## Referee Comment (RC2) · Anonymous Referee #2 · 8 Feb 2019

**1   General Comments**

The manuscript addresses the issue of how road construction impacts surface water/subsurface water flow in sloping fens. The authors present experimental as well as computational results on the investigation how different types of road constructions impact the flow dynamics particular with regard to negative implications like gully erosion.

The paper is well structured and written. In particular, the figures are meaningful and well prepared. The content is of scientific interest. However, I see potential for improvement, particular in the presentation and interpretation of simulation results. After

revising the manuscript according to the comments below, I see fit for a publication of the manuscript.

**2 Specific Comments**

**Section 1**

The author give a thorough literature review on the subject of road construction and its impact on flow, erosion and vegetation. The reader is well introduced to the topic of research and the motivation of the study. However, background information on the three road structures developed in Switzerland is missing (lines 90-93). Have there been more structures developed than those presented? What are advantages and disadvantages? Are there economical and/or constructional constrains to the choice of road types? Readers might benefit from (short) answers to these questions in the introduction or conclusion, understanding better the motivation for investigating the different road types.

**Section 2**

Section 2.2.1 + section 2.2.3:

Section 2.2 should be reworked. The authors use a well established subsurface-surface-water simulation software HGS where process equations are well known and documented. The equations (general processes) are given in the text, but the more relevant aspects of parameter choices and boundary/initial conditions (problem specific) are not or hardly discussed. Subsection 2.2.1 resembles a repetition of the HGS manual (e.g. the sentence in l. 197 on rivers and lakes is redundant). I fully agree with stating the relevant processes and naming the equations and parameters involved, but what is the benefit of giving the mathematical equations? Have they been modified in the code for the numerical study? The authors might consider cutting out the equations

**HESSD**

and giving proper reference to the used forms. Instead, the author should address all choices of model parameters. Give reference to Table 2. State the values of all input parameters (maybe additional table) and reason the choice and the source (measured values, educated guess, literature value etc.); e.g. explain the choice of the different Van-Genuchten parameters. Which are the most relevant parameters? Why is the sensitivity study chosen for the slope and K-values specifically? In total, the author should focus in this subsection on the core facts of the mathematics/physics behind and the relevant aspect for this specific case study. The authors should also give details on the choice of hydraulic conductivity values for the soil not only giving a reference (l. 235). The same for the values for the road drains (l.236) where there is not even a reference is given.

Figure 5 + section 2.2.2)

The resolution of the mesh cross sections in Figure 5 is rather low. It does not allow to identify any mesh structure. Specify the refinements made in the mesh (l.217). In figure 5c, are soil cells upstream connected with the soil cells below the road (not visible in figure with this resolution)? The mesh modifications for cases 5d, 5e and 5f show an artificial increase of inactive cells below the road (step shape instead of continuous slope form). This is not in line with the conceptual model structures given in Fig. 4. Shouldn't there be soil cells below the road construction? This might significantly modify the simulation results.

paragraph l.243-249

The text does not really refer to the sensitivity study but are more part of the model setup and analysis. To my opinion the locations of the observation points (Figure 6) are crucial for the interpretation of the different scenarios (see statement later). The author should clarify the coordinates of the observation points, particularly the distance to the road structures. The same holds for the observation depth. Are the velocities taken at a specific depth or are they depth averaged? Please specify in the text and

in Figure 6. I further recommend additional observation points. E.g. for comparison to flow velocities upstream, beneath the road structure and directly behind the road structure. Velocity profiles for the different road structures (and specific choices of parameter combinations) would be of interest.

**Section 3**

section 3.1 + Figure 7:

The resolution of the hydraulic head profiles should be adapted to the observed values in the first column for the sites SCH and STO, where the head profiles are not clearly observable in the current display form. The results for the EC contrasts (3rd column) are difficult to identify in the current form of presentation. I recommend a similar presentation as coloured pattern as in the 2nd column but preferably with a different colour scheme.

Section 3.2:

This section requires significant revision. The text is partially repetitive. Whereas several key aspects of the model results are not discussed and at some points explanation are missing.

paragraph l.288-292/Figures 8-10:

Skip figure 9 since the results presented are identical to figure 10 and therefore figure 9 is redundant. The y-axis should be labelled ($v$ in m/d) or at least it should be stated (with units) in the caption. There are artifacts in the middle column of Figure 8 (and 9) with circles and '+' in the label area of the y-axis.

paragraph l. 293 – 301:

The entire paragraph is repetitive and not to the point. Stick to the core message and argue with Darcy's law. I find the results for the flow velocities questionable. Or at least I see necessity for further analysis and discussion on the reported flow velocities. Lets

focus on the reference case without road construction and undisturbed flow. There are almost the same flow velocities reported for the KS1 and KS2 (Figure 8) although the soil conductivities are one order of magnitude different. The effect amplifies for increasing slope (Figure 10). Making a coarse estimate with Darcy's law (assuming constant gradient, full saturation and neglecting the effect of recharge, which is of course a simplification): $v = q/n = K/n\nabla(h)$. With a porosity of $n = 0.25$, $K = KS_1 = 8.64$ m/d and $\nabla h = 0.1$ (slope of $10\%$), we find $v = 3.456$ m/d. This value is more than one order of magnitude higher then the highest reported velocity of 0.274. Is this related to the surface runoff? There seems to be a upper flow velocity threshold of around 0.269 (l. 294, 303). Please explain and determine the general pattern for the flow dynamics.

paragraph l. 302-313:

The same as with the previous paragraph. Again an upper velocity threshold seems to be present. There seems also an apparent velocity threshold for the different drain conductivities (e.g. first column of figure 10). The explanation in l.309 – 313 is unsatisfying. Why are the results not comparable? I cannot see why flow velocities at the observation points should not be comparable for the grid adaption.

paragraph l. 314-324:

Again repetitive, not to the point, missing explanations. What is meant with "observed in the same transect". It is unclear to what the sentence in l. 318-319 refers to. Explain what is meant with "the difference along the transect is smaller" (l. 320). The message of the last sentence (l. 322-324) is unclear.

paragraph l. 325-335

The paragraph seems to repeat the arguments just stated in the previous paragraph. Thereby the numbers given are not identical (l. 333 compared to l. 319-320). In l. 333-335, the authors mention the effect of infiltration of low-conductivity soil layers, but it is not clearly displayed. Can infiltration above/through the road structure occur? Another

possible explanation: observed velocities depend on the distance of the observation points from the road structure. For very low hydraulic conductivities the flow dynamics downstream of the road have already formed similar to those upstream of the road. For high conductivities and thus high flow velocities the distance between the road and the observation points is not big enough to establish the previous flow pattern. Therefore the author should investigate additional observation points and provide velocity profiles (in x-direction) for the different road structures.

paragraph l. 336-347

The text is again repetitive, e.g. cut out sentence in l.339). The sentence in l. 345-346) does not make sense. The preferential pathways are not small-scale processes, they are subject to the heterogeneity of hydraulic conductivity. This can be resolved by continuum scale models, but not if assuming a spatially homogeneous conductivity. Furthermore, "the exact hydraulic head in an individual mini-piezometer" is not a process. I cannot agree with the sentence in l. 346-347; simulation results using a spatially homogeneous conductivity are not an average across preferential flow paths.

**3 Technical Corrections**

- l. 129: subsurface flows perpendicular→ subsurface flow is perpendicular

- l. 176: The mathematical representation of the nabla-operator is not fully correct. Please put the partial derivatives in brackets to symbolize its vector character.

- l. 176: modify formulation "with the outside of the simulation domain"

- l. 306 if the hydraulic conductivity → if the hydraulic soil conductivity

- l. 319: correct "from to 0.017"

- l. 367: rephrase to "both sides of the road where hydraulically connected for all investigated road structures"

- check references (particularly appearance and positions of doi's) as well as ref in l. 411

---

## Referee Comment (RC3) · Anonymous Referee #3 · 9 Feb 2019

This paper deals with a quite specific topic. The authors studied sloping fens and how their hydraulic processes/regimes are affected by roads. Referring to numerous studies, they nicely pointed out the importance of that topic. Despite a couple of typos, the majority of the manuscript is well written. However, I had the impression that the quality decreased a bit towards the end. At the beginning, the paper looked quite promising to me. After reading the entire manuscript, I was a bit disappointed as I expected a bit more. The field work and the modelling are totally decoupled. You somehow argue that a quantitative comparison is difficult due to preferential flow path, but couldn't it be realized by mapping and incorporating subsurface heterogeneities? A quantitative combination of your tracer tests and modelling, e.g. using results from

the tracer test for model calibration, would add some salt to the soup. With it you could prove the reliability of your model and subsequently you could run different adaptations (slopes etc.). Moreover, conclusions from the tracer test as well as model results are quite trivial (L-drains cause high flow velocities at their outlet and the higher the slope the stronger the effect). Actually, you don't need a model for such a conclusion. You raise a couple of interesting points, e.g. gully erosion and drying up of fens. Resulting questions are: What is the velocity threshold for gully erosion? How large is the area and to which extent fens downslope of the road are drying up in case of L-drains? Why don't you extent your story a bit towards these questions? From my point of view, the present manuscript lacks a bit of novelty and creativity. In order to meet the requirements for a journal like HESS, the story needs to be extended, e.g. regarding the above-mentioned ideas.

Below, you find a couple of minor comments:

General: Sometimes you are using spaces between numbers and operators and sometimes not. Please, check the guidelines of the journal.

Line 59: Capital "V" for Von Sengbusch. It's the start of a new sentence.

Figure 1: The cross-sectional view suggests that the water could easily pass underneath the road. However, in the text you mentioned that the top soil is very thin so that the road blocks the water flow to a large extent (also indicated by the lower figure). Isn't the figure a bit misleading? I would just increase a bit the size of the road and additionally sketch the impermeable bedrock.

Line 126: "similar" or "comparable" instead of "same" would be a more suitable word in this regard.

Line 131: I would add "bed" to "road bed structures".

Line 156: I wouldn't use the term "indirectly indicates". I would write something like "clearly shows". At least, I would skip "indirectly".

[Figure]

Line 157: Here, it is the other way around. Instead of writing "this indicates that there is no connection", I would be more careful by writing "this indicates a strongly hampered hydraulic connection".

Line 158: I would delete "and finally a decrease in EC is not expected". (It is just too obvious.)

Figure 3: For me, the cross sectional view is a bit superficial, but I guess this is a matter of taste… Still, the spaces before the question marks should be deleted. Moreover, I would just write "Piezometer" instead of "Mini-piezometer".

Line 163: What does "variable saturated" means? Sometimes saturated, sometimes unsaturated or variable hydraulic parameters? This should be explained more specific. (I guess it is a terminology from HGS.)

Line 166: I would replace "produce a sensitivity analysis and explore their sensitivities in" just by "analyse their impact on". Calling it sensitivity analysis is not really wrong, but for my taste not well fitting.

Section 2.2.1: I would strongly shorten this section, as it is not really a part of your story. If somebody is interested in the mathematics behind your model, he/she would read the original publication of HGS. I would write a couple of lines mentioning the basic assumptions and methods, but no equations. In case you really want to keep them, I have some minor suggestions:

(i) You should give the equations in the same order as referred to in the text, i.e. 1st Richard, 2nd Saint Venant, 3rd Darcy. Or just mention the diffusion a bit later in your text; (ii) Eq 1 and Eq 2 are modified versions of the Richards and Darcy. This should be mentioned. (iii) Line 176: No need to explain "Nabla". It's the common notation; (iv) Line 178: Commonly, "Uppercase Theta" is used for water content and not for porosity; (v) Line 180: I would add "saturated". K is the "saturated" hydraulic conductivity… (Multiplying with kr results in the actual hydraulic conductivity.)

Line 207f: "was used on the right face" – left and right are just a matter orientation. Maybe you better write something like: The lowest cells of the slope constitute a constant head boundary condition.

Line 218: Missing space between "2" and "m".

Line 234: Generally, I prefer the use of SI units, i.e. m/s instead of m/d.

Line 256f: What do you mean by "length scale of one to several meters". Is this a common expression?

Line 257: "629uS/am" – What is this? I guess 629 $\mu$S/cm...

Line 279: "local drying up of the soil)" – If you consider this as a problem, it would be quite easy to further investigate it with your numerical model. This would allow answering the question: how large is the affected area and to which extent it dries out?

Figure 7: In column 2 and 3 you are showing EC values. I am wondering why you are using totally different graphical representations. Moreover, if you are interpolating (I am not a big fan of interpolation, if it is not really necessary...), you should state which method you are using. What kind of background map you are using? Does it tell us something?

Line 288-292: For me, these lines are superficial. I would just delete them.

Line 293-301: This is very trivial and doesn't need any explanation. It can be directly derived from the Darcy equation (at least for the base case model).

Line 316f: Are you sure that "may be" is the right expression here?

Figure 8-10: It is not very comfortable to analyse the differences between the different slopes. Can't you just put all figures together using a slope specific colour?

Line 451: Is the year 2005 correct? I guess you want to refer to the manual, or? The one, I found, is from 2010.

---

## Editor Comment (EC1) · Anke Hildebrandt (Editor) · 20 Feb 2019

Dear authors,

three reviewers have provided feedback on the first version of your manuscript. The evaluations are really mixed. Some of the comments raised by reviewers #2 and #3 are rather critical not only regarding the presentation, but also challenging the content, such as the overall modeling strategy and novelty. You are encouraged to post a short comment during the interactive discussion phase, addressing the most critical general points. This provides a chance for direct interaction with the reviewers.

[Figure]

To allow for this interaction, I have extended the interactive discussion until end of February. Please also allow some time for the reviewers responses. There is no need yet to include a complete point to point response or new version of the manuscript, as there will be time to prepare and post it after the end of the interactive discussion.

I am looking forward to your comments, Anke Hildebrandt

———————————————

---

## Author Comment (AC1) · 25 Feb 2019

Dear all,

Firstly, we thank the three reviewers and the editor for their time and insightful comments on our manuscript. In general, the reviewers highlight the uniqueness of the study. In terms of novelty, we are not aware of any other study that has analyzed the perturbation through roads in wetlands in the way we have. Reviewer 2 appreciated the integration of modelling results with the experiment. A number of comments were provided, mainly related to the description of the modelling approach as well as the discussion and presentation of the results. We will carefully address these comments

and pay particular attention to the presentation of the results. We will carefully integrate these comments, see also below the response to reviewer 3. The reviewer also points to a number of technical aspects that need a more detailed justification and explanation. However, no fundamental technical issues were identified. Reviewer 3 provides a number of specific and constructive comments to improve the presentation and the interpretation of the results. A useful suggestion, for example, is concerning the interpretation of flow-velocities. We carefully will consider these suggestions and extend the post-processing of the results (new interpretations are already done and presented at the end of the document). As opposed to reviewer 1, reviewer 3 considers the fieldwork and the modelling decoupled. We can to a certain extent agree and believe that the link can be strengthened. However, a detailed reproduction through calibration of the field site is not considered to be useful as the results are entirely field-specific. The models provide a general framework for assessing the impact of roads on the flow of water. The additional postprocessing results requested by the reviewer 3 (described and listed below) will help to make this point stronger. In the revised manuscript, we will further provide detail on how the modelling results can be interpreted at the field site, and discuss with the consistency of the field data obtained. Specifically, the following results will be added:

1) A graph in which groundwater velocities are presented according to the slope, KS and KD to clearly identified which parameters govern the fen dynamics (Figure 1 below).

2) Analysis of groundwater velocities downslope the road at different distances to assess the extent of perturbation induced by the I-drain (Figure 2 below). In this way, the water distribution downgradient of the L-shape structure is addressed (as suggested by reviewer 3)

3) Analysis of groundwater velocities upslope the road as suggested by reviewer 2 (Figure 3 below). In this way, the impact of the road in the upstream part of the fen is assessed.
4) Other quantitative analyses according to reviewer comments, for example, the percentage of the area drained by the L-drain.

5) A comparison between simulated velocities and threshold velocity values above which gully erosion appears.

We will follow the good suggestions made by the reviewer 3 to improve and extend the discussion, which align also well with the comments of reviewer 2.

---

## Referee Comment (RC4) · Anonymous Referee #3 · 26 Feb 2019

Dear authors,

Your answer sounds quite promising and I am curious about reading the revision. If you would extent your story according to the listed points, I see potential for an improvement of your manuscript.

However, I still not really see the connection of the tracer test and modelling. I agree that a quantitative coupling (e.g. comparison of simulated and observed concentrations) will be very challenging caused by parameter heterogeneities, which are difficult to capture. Also, I can somehow agree to the argument that you want to provide a general modelling framework. However, this leaves me with the question: Why you

incorporate the tracer test at all? How does it support your synthetic model? Besides showing natural heterogeneities, you just prove that a L-drain constitutes a preferential flow path. Isn't that a bit too trivial?

Moreover, regarding the term novelty, we seem to have a slightly different opinion. For me novelty should be more than the application of an existing model to just a new case. Sure, not all HESS papers present an entirely new model or method, but they should present at least a creative solution or new combination of methods.

I encourage you to strongly revise your manuscript by adding some new ideas regarding e.g. drying up of fens or gully erosion (could be also something else). Basically, you should dig a bit deeper, but I am optimistic that you are able to do it.

---

## Referee Comment (RC5) · Anonymous Referee #2 · 27 Feb 2019

Dear authors,

I appreciate your quick comment and the efforts to include the reviewer suggestions. Based on the comment you gave, I see improvement in the revised manuscript. You nicely address many points raised by me and reviewer #3. However, after reading the other reviewer comments, I have to agree with Referee #3 that you should "dig a bit deeper" and present some novel results beyond the expected result that L-drains impact the strongest.

Thereby, I would like to encourage you to take fully advantage of your broad range of simulation scenarios. They allow to look at different road constructions from multiple

angles and different (parameter) settings. These results should be thoroughly analyzed and interpreted. In line with the my comments given in the first round of revisions and those of the 3rd Referee, I encourage you to define meaningful quantitative measures which help to provide inside into the system. These measures could be useful to make recommendations for road setup under different local settings. They might also be applied to the specific settings of the field tracer tests you presented to strengthen the link between the field and the numerical study.

---

## Author Comment (AC2) · 27 Mar 2019

Reviewer 2

**The author give a thorough literature review on the subject of road construction and its impact on flow, erosion and vegetation. The reader is well introduced to the topic of research and the motivation of the study.**

We thank the reviewer for the insightful comments.

**However, background information on the three road structures developed in Switzerland is missing (lines 90-93). Have there been more structures developed than those presented? What are advantages and disadvantages? Are there economical and/or constructional constrains to the choice of road types? Readers might benefit from (short) answers to these questions in the introduction or conclusion, understanding better the motivation for investigating the different road types.**

This suggestion is very useful and is integrated in the revision. We are not aware of any other structures. However, there are significant differences in the pricing for these road types. Information concerning this point will be added.

**Section 2**

**Section 2.2.1 + section 2.2.3:**

**Section 2.2 should be reworked. The authors use a well-established subsurface surface- water simulation software HGS where process equations are well known and documented. The equations (general processes) are given in the text, but the more relevant aspects of parameter choices and boundary/initial conditions (problem specific) are not or hardly discussed. Subsection 2.2.1 resembles a repetition of the HGS manual (e.g. the sentence in l. 197 on rivers and lakes is redundant). I fully agree with stating the relevant processes and naming the equations and parameters involved, but what is the benefit of giving the mathematical equations? Have they been modified in the code for the numerical study? The authors might consider cutting out the equations and giving proper reference to the used forms.**

Section 2.2.1 was completely reformulated as suggested. We have kept only the basic assumptions of HGS and gave references for a detailed HGS description, capabilities and application. The new subsection is presented below.

*The model used in the study is HydroGeoSphere (HGS) (Aquanty, 2017). HGS is a physically-based surface–subsurface fully-integrated model using the control volume finite element approach. HGS solves a modified Richards' equation describing the 3D subsurface flow. If the subsurface flow is not saturated, HGS employs the Van Genuchten (1980) functions to relate pressure head to saturation and relative hydraulic conductivity. Simultaneously, HGS also solves the 2D depth average diffusion-wave approximation of the Saint-Venant equation for describing the surface flow. To couple surface and subsurface and simulate the water exchanges between both domains, the "dual node approach" is used. In this approach, the top nodes representing the ground surface are used for calculating both subsurface and surface flow. The water exchanges are calculated as hydraulic head differences of the two domains and multiplied by the vertical hydraulic conductivity of the top layer and a coupling factor.*

*The iterative Newton-Raphson method is used to solve the nonlinear equations. At each subsurface node, saturation and groundwater heads are calculated, which allows for the calculation of the Darcy flux. On the surface domain, the surface water heights are calculated at each node to determine surface water flux. Rivers and lakes are characterized by a surface water depth larger than 0. For further details on the code, HGS capabilities and application, see Aquanty (2017), Brunner and Simmons (2012) or Cochand et al. (2019).*

**Instead, the author should address all choices of model parameters. Give reference to Table 2. State the values of all input parameters (maybe additional table) and reason the choice and the source (measured values, educated guess, literature value etc.); e.g. explain the choice of the different Van-Genuchten parameters. Which are the most relevant parameters? Why is the sensitivity study chosen for the slope and K-values specifically? In total, the author should focus in this subsection on the core facts of the mathematics/physics behind and the relevant aspect for this specific case study. The authors should also give details on the choice of hydraulic conductivity values for the soil not only giving a reference (l. 235). The same for the values for the road drains (l.236) where there is not even a reference is given.**

We agree with these comments, they will certainly help to clarify the manuscript. Regarding a more detailed model parameter descriptions, section 2.2.3. You find below the suggested reworked section 2.2.3

[revised manuscript text omitted]

**Figure 5 + section 2.2.2**

**The resolution of the mesh cross sections in Figure 5 is rather low. It does not allow to identify any mesh structure. Specify the refinements made in the mesh (l.217).**

The size of Figure 5 was increased (see below). Now the discretization is well represented in figure 5b to 5f. Unfortunately, the size of the figure should be much bigger (about A3) to see clearly the mesh refinement… Therefore we added discretization details in figure 5a to inform the reader.

[Figure]

**In figure 5c, are soil cells upstream connected with the soil cells below the road (not visible in figure with this resolution)?**

Yes in figure5c soil cells are connected. With the modification of figure5, now the connection can be seen

**The mesh modifications for cases 5d, 5e and 5f show an artificial increase of inactive cells below the road (step shape instead of continuous slope form). Shouldn't there be soil cells below the road construction? This might significantly modify the simulation results.**

When a road construction takes place, impermeable material is excavated upstream and filled downstream (see below). In order to implement this engineering structure in the model, inactive cells need to be present below the road. This conceptualization is therefore consistent with the construction of these road-types.

**This is not in line with the conceptual model structures given in Fig. 4.**

Yes, it is true it is not in line with the figure4. Therefore, we modified it as presented below.

[Figure]

1. Soil layer 0.40 m

2. Road - concrete layer 2.50 x 0.15 and compact sand 2.50 x 0.20 m

3. Road - compact gravel-clay 2.0 x 0.35 m

4. Low density gravel

5. Drain with coarse gravel

6. Log of wood and free space

7. Fixed GW head BC egal to 2m on all the face

8. Critical depth BC egal to ground surface elevation (2m) on top nodes

9. Fixed constant flux equal to 380 mm/y

Impermeable layer (inactive cells)

**Paragraph l.243-249**

**The text does not really refer to the sensitivity study but are more part of the model setup and analysis.**

We agree, the sensitivity analysis is a part of model setup and analysis. Therefore, we changed the name of this paragraph "model setup".

**To my opinion the locations of the observation points (Figure 6) are crucial for the interpretation of the different scenarios (see statement later). The author should clarify the coordinates of the observation points, particularly the distance to the road structures.**

We also agree, the location of the observation points (now sections) is crucial. We modified Figure 6 accordingly , and added the distance the observation points and the road.

[Figure]

**The same holds for the observation depth. Are the velocities taken at a specific depth or are they depth averaged? Please specify in the text and in Figure 6. I further recommend additional observation points. E.g. for comparison to flow velocities upstream, beneath the road structure and directly behind the road structure. Velocity profiles for the different road structures (and specific choices of parameter combinations) would be of interest.**

Instead of extract velocities, it would be clearer to extract the subsurface flow rate through a section. Therefore we suggest extracting flow rate through 1m wide sections in the soil layer located upstream and downstream the road as presented in figure 6. Therefore all figures were modified (from velocities to flow rates).

I addition, the following results are presented:

1) Analysis of groundwater flow rates upslope the road (Figure 1). In this way, the impact of the road in the upstream part of the fen is assessed.
2) Analysis of groundwater flow rates downslope the road at different distances to assess the extent of perturbation induced by the l-drain (Figure 2). In this way, the water distribution downgradient of the L-shape structure is addressed.
3) A graph in which flow rates are presented according to the slope, KS and KD to clearly identified which parameters govern the fen dynamics (Figure 3)

[Figure]

**Figure 1 : Simulated groundwater velocities 2.5 m upstream each road structures and each parameter combination with a slope of a) 10%, b) 20% and c) 30%.**

[Figure]

**Figure 2 : Extent of perturbations due to the l-drain road type: Simulated groundwater flow rates at different distances of the road.**

[Figure]

**Figure 3 : Simulated groundwater velocities at observation point G depending on the slope, KS and KD**

**Section 3**

This section will be reworked to be less repetitive as you mention. In addition, a new section will be added to assess the potential risk of gully erosion. To do that, the simulated groundwater flow rate will be compared with the maximum flux than can flow in the soil calculated with the Darcy law. If the road structure induces a groundwater flow higher than the soil capacity then gully may occur. For example in the surrounded plot in Figure 4, you see that L-drain induces a groundwater flow rate higher than the soil capacity and therefore may induce gully erosion.

[Figure]

**Figure 4 : Simulated groundwater velocities 2 m downstream each road structures and each parameter combination with a slope 20%.**

**section 3.1 + Figure 7:**

**The resolution of the hydraulic head profiles should be adapted to the observed values in the first column for the sites SCH and STO, where the head profiles are not clearly observable in the current display form. The results for the EC contrasts (3rd column) are difficult to identify in the current form of presentation. I recommend a similar presentation as coloured pattern as in the 2nd column but preferably with a different colour scheme.**

The modification will be done according to your comments.

**Section 3.2:**
**This section requires significant revision. The text is partially repetitive. Whereas several key aspects of the model results are not discussed and at some points explanation are missing.**

**paragraph l. 293 – 301:**

**The entire paragraph is repetitive and not to the point. Stick to the core message and argue with Darcy's law. I find the results for the flow velocities questionable. Or at least I see necessity for further analysis and discussion on the reported flow velocities. Lets focus on the reference case without road construction and undisturbed flow. There are almost the same flow velocities**

**reported for the KS1 and KS2 (Figure 8) although the soil conductivities are one order of magnitude different. The effect amplifies for increasing slope (Figure 10). Making a coarse estimate with Darcy's law (assuming constant gradient, full saturation and neglecting the effect of recharge, which is of course a simplification): v = q=n = K=nr(h). With a porosity of n = 0:25, K = KS1 = 8:64 m/d and rh = 0:1 (slope of 10%), we find v = 3:456 m/d. This value is more than one order of magnitude higher then the highest reported velocity of 0.274. Is this related to the surface runoff? There seems to be a upper flow velocity threshold of around 0.269 (l.294, 303). Please explain and determine the general pattern for the flow dynamics.**

In the base case and all others models, the precipitation is 380mm/year. It means that at x=65.5m in the model, the maximum flow rates with this precipitation rate is:

$$Q = 65.5\ (m)\ \times\ 380\ (mm/y)\ \times\ \frac{1}{1000}\ (m/mm)\ \times\ \frac{1}{365}\ (y/d)\ =\ 0.068 m3/d/m$$

The maximum flow rate according to the soil KS1 (8.64) and a slope of 10% is:

$$Q =\ q \times A = Ks \times \nabla H \times A = 8.6 \times 0.1 \times 0.4\ \times 1 = 0.345 m3/d$$

It means that the maximum flow rate in the soil may be more important than precipitation. It is however not always the case in the other models. In the new analysis of model results, we will compare the simulated flow rate vs. the maximum of flow rate of the soil to see if the simulate flow rate is close to the maximum of the soil. We will also compare the simulated flow rates and the maximum flow rates due to the precipitation (as previously calculated) to assess more in detail the concentration of the flux induced by road structures.

**paragraph l. 302-313**
**The same as with the previous paragraph. Again an upper velocity threshold seems to be present. There seems also an apparent velocity threshold for the different drain conductivities (e.g. first column of figure 10). The explanation in l.309 – 313 is unsatisfying. Why are the results not comparable? I cannot see why flow velocities at the observation points should not be comparable for the grid adaption.**

The threshold is due to precipitation rate which limits the flow rate in the subsurface.

In the figure below, you see the mesh of the no-excavation model. It was impossible to develop the model without a small extension of the road and drains in the soil layer because of the mesh geometry. This extension is surrounded in red in the figure. The extension induced artefact in results. Therefore we decided not to include these results. In 20% and 30% slope models, the slope is steep enough to develop the model without this extension.

[Figure]

**paragraph l. 314-324:**
**Again repetitive, not to the point, missing explanations. What is meant with "observed in the same transect". It is unclear to what the sentence in l. 318-319 refers to. Explain what is meant with "the difference along the transect is smaller" (l. 320). The message of the last sentence (l. 322-324) is unclear.**

"observed in the same transect" means observed along the transect formed by the observation section A, B, C, D, E, F, G and H. In other words, it means the simulated flow rates downslope the road in a same model.

"the difference along the transect is smaller" means that difference between G and C observation sections is smaller in a specific model than in another.

For the line 322-324, we wanted to say that the slope increases the differences between maximum and minimum simulated flow rates downslope the road.

This paragraph will be reworked because it is not very clear as you mention. Another word will used instead of transect to describe simulated flow rate downstream the road.

**paragraph l. 325-335**
**The paragraph seems to repeat the arguments just stated in the previous paragraph. Thereby the numbers given are not identical (l. 333 compared to l. 319-320). In l. 333-335, the authors mention the effect of infiltration of low-conductivity soil layers, but it is not clearly displayed. Can infiltration above/through the road structure occur?**

Drains located along the road act like an infiltration drain because it drains a part of the runoff water.

**Another possible explanation: observed velocities depend on the distance of the observation points from the road structure. For very low hydraulic conductivities the flow dynamics downstream of the road have already formed similar to those upstream of the road. For high conductivities and thus high flow velocities the distance between the road and the observation points is not big enough to establish the previous flow pattern. Therefore the author should investigate additional observation points and provide velocity profiles (in x-direction) for the different road structures.**

We agree that a profile in x direction may be usefull to have a better understanding on the dynamics. In the interests of brevity we suggest to create this profile for cases in which the flow rate is increased (when the soil layer = KS3).

**paragraph l. 336-347**
**The text is again repetitive, e.g. cut out sentence in l.339). The sentence in l. 345- 346) does not make sense. The preferential pathways are not small-scale processes, they are subject to the heterogeneity of hydraulic conductivity. This can be resolved by continuum scale models, but not if assuming a spatially homogeneous conductivity. Furthermore, "the exact hydraulic head in an individual mini-piezometer" is not a process. I cannot agree with the sentence in l. 346-347; simulation results using a spatially homogeneous conductivity are not an average across preferential flow paths.**

This section will also reworked to make it less repetitive and sentence l 345-346 will be clarified. We also agree that "hydraulic head" is not a process. "Processes" will be remplaced by "observations". Clearly an average hydraulic conductivity cannot represent the dynamics in individual flow paths but may represent the average dynamics of multiple flow path and less conductive parts. We will reformulate the sentence accordingly.

**Technical corrections:**

**l. 129: subsurface flows perpendicular -> subsurface flow is perpendicular**
Yes, we corrected "subsurface flows perpendicular" by "subsurface is perpendicular". The corrected sentence is: …another important criterion for the selection of the study areas was that subsurface flow is perpendicular to the road.

**l. 176: The mathematical representation of the nabla-operator is not fully correct. Please put the partial derivatives in brackets to symbolize its vector character.**

These lines were removed

**l. 176: modify formulation "with the outside of the simulation domain"**

These lines were removed

**l. 306 if the hydraulic conductivity -> if the hydraulic soil conductivity**

Yes you are true, it is clearer if add "soil".

**l. 319: correct "from to 0.017"**

Yes it is a mistake. We removed the useless "to".

**l. 367: rephrase to "both sides of the road where hydraulically connected for all investigated road structures"**

Yes, we corrected, the sentence is "The tracer tests showed that both sides of the road where hydraulically connected for all investigated road structures."

**check references (particularly appearance and positions of doi's) as well as ref in l. 411**

We checked the reference (Deroze 1998), the doi is unusual but it is correct.

discretization

---

## Author Comment (AC3) · 27 Mar 2019

**Questions and comments of the reviewer 3 are in bold**

**Your answer sounds quite promising and I am curious about reading the revision. If you would extent your story according to the listed points, I see potential for an improvement of your manuscript.**

Thank you for the comment.

**However, I still not really see the connection of the tracer test and modelling. I agree that a quantitative coupling (e.g. comparison of simulated and observed concentrations) will be very challenging caused by parameter heterogeneities, which are difficult to capture. Also, I can somehow agree to the argument that you want to provide a general modelling framework. However, this leaves me with the question: Why you Discussion paper incorporate the tracer test at all? How does it support your synthetic model? Besides showing natural heterogeneities, you just prove that a L-drain constitutes a preferential flow path. Isn't that a bit too trivial?**

In our experience it never hurts to have field experiments backing up a modelling approach—actually quite the opposite.  Even if it might appear trivial at a first glance we believe there is always value in the field data.  Apart from this general consideration, we don't think it is a trivial as mentioned by the reviewer. Below some examples (which we will elaborate in the revised manuscript):

- It is also not all clear how important the natural heterogeneities and preferential pathways are in comparison with the drain.
- The price differences of these engineering structures are significant. Given that the models always need to simplify a system it is in our experience unwise to base decision purely on modelling approaches--- the most convincing approach is a combination of both with a demonstration that the planned systems work as planned, and then the models can help to identify how the proposed system will affect flow under different conditions.
- It could also  be that the engineering structure is not well implemented or has not been communicated correctly. This is in fact a very common problem. It is not at all trivial to implement these engineering structures in wetland, as the construction machines cannot leave the road,  access is difficult and there are legal considerations ect. With the field test we show that these structures can be built and functions as planned.

We want to highlight that this paper is directed not only towards the scientific community, but also stakeholder and the engineering firms who implement these structures. It it therefore particularly important to demonstrate that model can reproduce the general behaviour observed in the field.

Finally, it is clear that below the road and immediately downgradient, the L-structure create a preferential flow path. However, the key question is how quickly flow redistributes again laterally further downgradient. With the suggested modifications (described in the next section), this question can be answered.

**Moreover, regarding the term novelty, we seem to have a slightly different opinion. For me novelty should be more than the application of an existing model to just a new case. Sure, not all HESS papers present an entirely new model or method, but they should present at least a creative solution or new combination of methods. I encourage you to strongly revise your manuscript by adding some new ideas regarding e.g. drying up of fens or gully erosion (could be also something else). Basically, you should dig a bit deeper, but I am optimistic that you are able to do it.**

In term of novelty we added a range of points as suggested by the reviewer. We agree that more results can be extracted from the modelling approach. It is the first time this topic is treated, and we also want to highlight that physically based models such as the ones we use are not that commonly used. Finally, we want to highlight that HESS also encourages the submission of applied research, as highlighted in the description of the journal:

*"HESS encourages and supports fundamental and applied research that advances the understanding of hydrological systems, their role in providing water for ecosystems and society, and the role of the water cycle in the functioning of the Earth system. "*

In addition, section 3 will be reworked and new subsection will be added in which we assess the potential risk of gully erosion. To achieve this, the simulated groundwater flow rate will be compared with the maximum flux than can flow in the soil calculated with the Darcy law. If the road structure induces a groundwater flow higher than the soil capacity then gully may occur. For example in the surrounded plot in Figure below, you see that L-drain induces a groundwater flow rate higher than the soil capacity and therefore may induce gully erosion.

[Figure]

**Simulated groundwater velocities 2 m downstream each road structures and each parameter combination with a slope 20%.**

Finally, the simulation results will at different distance of the road to have a better assessment of the road impact. We will be also able to identify areas in which the soil layer is not fully saturated or on the contrary areas in which runoff occurs. See an example in the figure below.

[Figure]

**Extent of perturbations due to the l-drain road type: Simulated groundwater flow rates at different distances of the road.**

**Minor comments**

**General: Sometimes you are using spaces between numbers and operators and sometimes not. Please, check the guidelines of the journal.**

Spaces between numbers and units were removed as described in the guideline.

**Line 59: Capital "V" for Von Sengbusch. It's the start of a new sentence.**

Capital V was corrected.

**Figure 1: The cross-sectional view suggests that the water could easily pass underneath the road. However, in the text you mentioned that the top soil is very thin so that the road blocks the water flow to a large extent (also indicated by the lower figure). Isn't the figure a bit misleading? I would just increase a bit the size of the road and additionally sketch the impermeable bedrock.**

The size of the road in the figure 1 was increased and impermeable bedrock was added. In this way, the reader will directly understand that the groundwater is blocked upstream the road.

[Figure]

**Line 126: "similar" or "comparable" instead of "same" would be a more suitable word in this regard.**

"same" was replaced by "similar".

**Line 131: I would add "bed" to "road bed structures"**

"Bed" was added, now the sentence is: To evaluate the hydraulic connection provided by the road bed structures, tracer tests were carried out.

**Line 156: I wouldn't use the term "indirectly indicates". I would write something like "clearly shows". At least, I would skip "indirectly".**

The term "indirectly indicates" was removed and replaced by "clearly shows". Now the sentence is: An increase in EC in piezometers located in the downslope area indicates that the injected salt water flowed from the upslope area to the downslope area below the road and clearly shows a hydraulic connection.

**Line 157: Here, it is the other way around. Instead of writing "this indicates that there is no connection", I would be more careful by writing "this indicates a strongly hampered hydraulic connection".**

Yes it is more finely described if we use "strongly hampered" instead of "no connection". We also removed "finally a decrease in EC is not expected". After correction, the whole sentence is: Conversely, if no changes in EC are observed in piezometers, this indicates a strongly hampered hydraulic connection below the road.

**Line 158: I would delete "and finally a decrease in EC is not expected". (It is just too obvious.)**

We can remove this line if you think that it is too obvious.

**Figure 3: For me, the cross sectional view is a bit superficial, but I guess this is a matter of taste: Still, the spaces before the question marks should be deleted. Moreover, I would just write "Piezometer" instead of "Mini-piezometer".**

The figure 3 was modified according to your comments.

[Figure]

**Line 163: What does "variable saturated" means? Sometimes saturated, sometimes unsaturated or variable hydraulic parameters? This should be explained more specific (I guess it is a terminology from HGS.)**

Variably saturated means that change in saturation of the soil is simulated. However, is not important to mention that here. To be clearer, we changed "variably saturated subsurface water flow" by "subsurface water flow". The corrected sentence is: First, a 3D base case model representing surface and subsurface water flow in a sloping fen was elaborated.

**Line 166: I would replace "produce a sensitivity analysis and explore their sensitivities in" just by "analyse their impact on". Calling it sensitivity analysis is not really wrong, but for my taste not well fitting.**

In our opinion, it is a sensitivity analysis however, we can change. The suggested would be:

*For each model, various slopes, organic soil and road drain hydraulic conductivities were implemented to produce a sensitivity analysis and analyse their impact on the sloping fen flow dynamics*

**Section 2.2.1: I would strongly shorten this section, as it is not really a part of your story. If somebody is interested in the mathematics behind your model, he/she would read the original publication of HGS. I would write a couple of lines mentioning the basic assumptions and methods, but no equations. In case you really want to keep them, I have some minor suggestions:**

Yes you are true section may be reduced (reviewer 2 made the same comment). We keep only the main assumptions and method.

**(i) You should give the equations in the same order as referred to in the text, i.e. 1st Richard, 2nd Saint Venant, 3rd Darcy. Or just mention the diffusion a bit later in your text; (ii) Eq 1 and Eq 2 are modified versions of the Richards and Darcy. This should be mentioned. (iii) Line 176: No need to explain "Nabla". It's the common notation; (iv)Line 178: Commonly, "Uppercase Theta" is used for water content and not for porosity;(v) Line 180: I would add "saturated". K is the "saturated" hydraulic conductivity:(Multiplying with kr results in the actual hydraulic conductivity.)**

These lines were removed.

**Line 207f: "was used on the right face" – left and right are just a matter orientation. Maybe you better write something like: The lowest cells of the slope constitute a constant head boundary condition.**

Yes, it is better to use "the lowest cells of the slope" than "one the right face". After correction the sentence is: A constant groundwater head boundary condition (Dirichlet type) equal to the ground

surface elevation (2m) was used on the lowest cells of the slope (x=76m on the **Erreur ! Source du renvoi introuvable.**a) allowing the groundwater to flow out of the model

**Line 218: Missing space between "2" and "m".**

According to the guideline, we should not use a space between a number and an abbreviation of a unit. Therefore, we removed all spaces in the manuscript.

**Line 234: Generally, I prefer the use of SI units, i.e. m/s instead of m/d.**

As hydrogeologist, we also prefer m/s instead of m/d, however, the manuscript is not only hydrogeologist but for other environmental sciences such biologists. In my opinion, m/d provides greater clarity.

**Line 256f: What do you mean by "length scale of one to several meters". Is this a common expression?**

"length scale of one to several meters" is not a common expression but a mistake. We removed "length scale". Now the sentence is: In contrast, the EC maps established prior to the tracer test show a spatial variability of one to several meters

**Line 257: "629uS/am" – What is this? I guess 629 _S/cm**

Yes is 629μS/am. We corrected it. Thank you for carefully reading our paper!

**Line 279: "local drying up of the soil)" – If you consider this as a problem, it would be quite easy to further investigate it with your numerical model. This would allow answering the question: how large is the affected area and to which extent it dries out?**

Yes you are absolutely right. Therefore, a new figure (figure 9) was created in which we can see the extent of perturbations induced be the l-drain structure.

**Figure 7: In column 2 and 3 you are showing EC values. I am wondering why you are using totally different graphical representations. Moreover, if you are interpolating (I am not a big fan of interpolation, if it is not really necessary: : :), you should state which method you are using. What kind of background map you are using? Does it tell us something?**

This figure will be corrected and the background and the interpolation method will be specified.

**Line 288-292: For me, these lines are superficial. I would just delete them.**

We wanted to help the reader by describing each step of the result interpretation. If you think it is superficial, we can remove them.

**Line 293-301: This is very trivial and doesn't need any explanation. It can be directly derived from the Darcy equation (at least for the base case model).**

Yes it is trivial it can be directly derived from the Darcy equation. However, it seems important to describe the base case insofar as the base case is used to compare other results.

**Line 316f: Are you sure that "may be" is the right expression here?**

We modified the "may be" by "can be".

**Figure 8-10: It is not very comfortable to analyse the differences between the different slopes. Can't you just put all figures together using a slope specific colour?**

Yes it is true. We grouped together the three slopes.

**Line 451: Is the year 2005 correct? I guess you want to refer to the manual, or? The one, I found, is from 2010.**

Yes it was the former version. However we should use the new reference: Aquanty: HydroGeoSphere, a three-dimensional numerical model describing fully- integrated subsurface and surface flow and solute transport. Waterloo, ON, Canada., 2017.

---

## Author Comment (AC4) · 1 Apr 2019

We thank you very much for the time you've put into the review of our article and the very positive comments. Best regards.

---

## Author Comment (AC5) · 2 Apr 2019

We thank you very much for the time you've put into the review of our article. The responses of your two comments RC2 and RC5 can be found in the answer AC2. Thank you.

---

## Author Comment (AC6) · 2 Apr 2019

We thank you very much for the time you've put into the review of our article. The responses of your two comments RC3 and RC4 can be found in the answer AC3. Thank you.

---

## Author Response (AR1)

**Dear editor,**

**Please find in this document our answers.**

**We thank you for the time you devoted for our paper review.**

**The authors.**

**Reviewer 1**

**We thank you for your very positive comments.**

**Reviewer 2**

**We thank you for your time and the pertinent comments and questions. Please find below our answers and the modified manuscript.**

**The author give a thorough literature review on the subject of road construction and its impact on flow, erosion and vegetation. The reader is well introduced to the topic of research and the motivation of the study.**

We thank the reviewer for the insightful comments.

**However, background information on the three road structures developed in Switzerland is missing (lines 90-93). Have there been more structures developed than those presented? What are advantages and disadvantages? Are there economical and/or constructional constrains to the choice of road types? Readers might benefit from (short) answers to these questions in the introduction or conclusion, understanding better the motivation for investigating the different road types.**

This suggestion is very useful and is integrated in the revision. We are not aware of any other structures. However, there are significant differences in the pricing for these road types. Unfortunatly we did not receive this information yet. This point will be added as soon as we have the information.

Section 2

Section 2.2.1 + section 2.2.3:

**Section 2.2 should be reworked. The authors use a well-established subsurface surface- water simulation software HGS where process equations are well known and documented. The equations (general processes) are given in the text, but the more relevant aspects of parameter choices and boundary/initial conditions (problem specific) are not or hardly discussed. Subsection 2.2.1 resembles a repetition of the HGS manual (e.g. the sentence in l. 197 on rivers and lakes is redundant). I fully agree with stating the relevant processes and naming the equations and parameters involved, but what is the benefit of giving the mathematical equations? Have they been modified in the code for the numerical study? The authors might consider cutting out the equations and giving proper reference to the used forms.**

Section 2.2.1 was completely reformulated as suggested. We have kept only the basic assumptions of HGS and gave references for a detailed HGS description, capabilities and application. The new subsection is presented below in 178 to 188.

**Instead, the author should address all choices of model parameters. Give reference to Table 2. State the values of all input parameters (maybe additional table) and reason the choice and the source (measured values, educated guess, literature value etc.); e.g. explain the choice of the different Van-Genuchten parameters. Which are the most relevant parameters? Why is the sensitivity study chosen for the slope and K-values specifically? In total, the author should focus in this subsection on the core facts of the mathematics/physics behind and the relevant aspect for**

this specific case study. The authors should also give details on the choice of hydraulic conductivity values for the soil not only giving a reference (l. 235). The same for the values for the road drains (l.236) where there is not even a reference is given.

We agree with these comments and section 2.2.3 reworked (see line 202 to 232)

**Figure 5 + section 2.2.2**

**The resolution of the mesh cross sections in Figure 5 is rather low. It does not allow to identify any mesh structure. Specify the refinements made in the mesh (l.217).**

The size of Figure 5 was increased (see below). Now the discretization is well represented in figure 5b to 5f. Unfortunately, the size of the figure should be much bigger (about A3) to see clearly the mesh refinement... Therefore we added discretization details in figure 5a to inform the reader. You find the new figure in line 198.

[Figure]

**In figure 5c, are soil cells upstream connected with the soil cells below the road (not visible in figure with this resolution)?**

Yes in figure5c soil cells are connected. With the modification of figure5, now the connection can be seen

**The mesh modifications for cases 5d, 5e and 5f show an artificial increase of inactive cells below the road (step shape instead of continuous slope form). Shouldn't there be soil cells below the road construction? This might significantly modify the simulation results.**

When a road construction takes place, impermeable material is excavated upstream and filled downstream (see below). In order to implement this engineering structure in the model, inactive cells need to be present below the road. This conceptualization is therefore consistent with the construction of these road-types.

**This is not in line with the conceptual model structures given in Fig. 4.**

Yes, it is true it is not in line with the figure4. Therefore, we modified it as presented below. You find the modified figure in line 183

[Figure]

Soil layer 0.40 m

Road - concrete layer 2.50 x 0.15 and compact sand 2.50 x 0.20 m

Road - compact gravel-clay 2.0 x 0.35 m

Low density gravel

Drain with coarse gravel

Log of wood and free space

Fixed GW head BC egal to 2m on all the face

Critical depth BC egal to ground surface elevation (2m) on top nodes

Fixed constant flux equal to 380 mm/y

+ Impermeable layer (inactive cells)

a) Base case
b) No-excavation
c) L-drain
d) Wood-log

**Paragraph l.243-249**

**The text does not really refer to the sensitivity study but are more part of the model setup and analysis.**

We agree, the sensitivity analysis is a part of model setup and analysis. Therefore, we changed the name of this paragraph "model setup" (see line 202).

**To my opinion the locations of the observation points (Figure 6) are crucial for the interpretation of the different scenarios (see statement later). The author should clarify the coordinates of the observation points, particularly the distance to the road structures.**

We also agree, the location of the observation points (now sections) is crucial. We modified Figure 6 accordingly , and added the distance the observation points and the road (see line 233)

[Figure]

**The same holds for the observation depth. Are the velocities taken at a specific depth or are they depth averaged? Please specify in the text and in Figure 6. I further recommend additional observation points. E.g. for comparison to flow velocities upstream, beneath the road structure and directly behind the road structure. Velocity profiles for the different road structures (and specific choices of parameter combinations) would be of interest.**

Instead of extract velocities, it would be clearer to extract the subsurface flow rate through a section. Therefore we suggest extracting flow rate through 1m wide sections in the soil layer located upstream and downstream the road as presented in figure 6. Therefore all figures were modified (from velocities to flow rates).

I addition, the following results are presented:

1) Analysis of groundwater flow rates upslope the road (now figure 12 line 394 and presented below). In this way, the impact of the road in the upstream part of the fen is assessed.
2) Analysis of groundwater flow rates downslope the road at different distances to assess the extent of perturbation induced by the I-drain (now figure 10 line 389 and presented below). In this way, the water distribution downgradient of the L-shape structure is addressed.
3) Surface flow concentration caused by the L-drain (now figure 11 line 392 and presented below)

[Figure]

**Figure 11: Simulated groundwater flow rates 2.5m upstream each road structures and each parameter combination with a slope of a) 10%, b) 20% and c) 30%.**

[Figure]

**Soil hydraulic conductivity KS1= 8.64 m/d Slope 10 %** — **Soil hydraulic conductivity KS1= 8.64 m/d Slope 20 %** — **Soil hydraulic conductivity KS1= 8.64 m/d Slope 30 %**

Drain hydraulic conductivity KD1 = 8640 m/d
Drain hydraulic conductivity KD2 = 864 m/d
Drain hydraulic conductivity KD3 = 86.4 m/d

Groundwater flow rate (m³/d)

Legend: + no road    ◇ 2.5m upstream    □ 2m downstream    △ 3.5m downstream    + 6.5m downstream

Figure 9 : Extent of perturbations due to the l-drain road type: Simulated groundwater v flow rates at G section at different distances the road.

**Figure 10: Simulated surface flow of the KS2-KD2 model and a slope of 20% for each road structure**

**Section 3**

This section (mainly 3.2 line 271) was reworked to be less repetitive as you mention. In addition, a paragraph was added to assess the potential risk of gully erosion. To do that, the simulated groundwater flow rate will be compared with the maximum flux than can flow in the soil calculated with the Darcy law. If the road structure induces a groundwater flow higher than the soil capacity then gully may occur. For example in the surrounded plot in Figure 4, you see that L-drain induces a groundwater flow rate higher than the soil capacity and therefore may induce gully erosion.

[Figure]

**Figure 1 : Simulated groundwater flow rate 2 m downstream each road structures and each parameter combination with a slope 20%.**

**section 3.1 + Figure 7:**

**The resolution of the hydraulic head profiles should be adapted to the observed values in the first column for the sites SCH and STO, where the head profiles are not clearly observable in the current display form. The results for the EC contrasts (3rd column) are difficult to identify in the current form of presentation. I recommend a similar presentation as coloured pattern as in the 2nd column but preferably with a different colour scheme.**

We try to modify the third column as you mention but in our opinion, it is not a better representation of results (see the figure below). What do you think ?

[Figure]

For the resolution, do you mean the resolution of the picture or the intervals between level lines ?

**Section 3.2:**
**This section requires significant revision. The text is partially repetitive. Whereas several key aspects of the model results are not discussed and at some points explanation are missing.**

The section 3.2 was reworked (see line 271 to 372) and below your find our detailed answers to your relevant comments or questions.

**paragraph l. 293 – 301:**

**The entire paragraph is repetitive and not to the point. Stick to the core message and argue with Darcy's law. I find the results for the flow velocities questionable. Or at least I see necessity for further analysis and discussion on the reported flow velocities. Lets focus on the reference case without road construction and undisturbed flow. There are almost the same flow velocities reported for the KS1 and KS2 (Figure 8) although the soil conductivities are one order of magnitude different. The effect amplifies for increasing slope (Figure 10). Making a coarse estimate with Darcy's law (assuming constant gradient, full saturation and neglecting the effect of recharge, which is of course a simplification): v = q=n = K=nr(h). With a porosity of n = 0:25, K = KS1 = 8:64 m/d and rh = 0:1 (slope of 10%), we find v = 3:456 m/d. This value is more than one order of magnitude higher then the highest reported velocity of 0.274. Is this related to the surface runoff? There seems to be a upper flow velocity threshold of around 0.269 (l.294, 303). Please explain and determine the general pattern for the flow dynamics.**

In the base case and all others models, the precipitation is 380mm/year. It means that at x=65.5m in the model, the maximum flow rates with this precipitation rate is:

$$Q = 65.5 \, (m) \, \times \, 380 \, (mm/y) \, \times \, \frac{1}{1000} \, (m/mm) \, \times \, \frac{1}{365} \, (y/d) \, = \, 0.068 m3/d/m$$

The maximum flow rate according to the soil KS1 (8.64) and a slope of 10% is:

$$Q = \, q \times A = Ks \times \nabla H \times A = 8.6 \times 0.1 \times 0.4 \, \times 1 = 0.345 m3/d$$

It means that the maximum flow rate in the soil may be more important than precipitation. It is however not always the case in the other models. In the new analysis of model results, we will compare the simulated flow rate vs. the maximum of flow rate of the soil to see if the simulate flow rate is close to the maximum of the soil. We will also compare the simulated flow rates and the maximum flow rates due to the precipitation (as previously calculated) to assess more in detail the concentration of the flux induced by road structures.

**paragraph l. 302-313**
**The same as with the previous paragraph. Again an upper velocity threshold seems to be present. There seems also an apparent velocity threshold for the different drain conductivities (e.g. first column of figure 10). The explanation in l.309 – 313 is unsatisfying. Why are the results not comparable? I cannot see why flow velocities at the observation points should not be comparable for the grid adaption.**

The threshold is due to precipitation rate which limits the flow rate in the subsurface.

In the figure below, you see the mesh of the no-excavation model. It was impossible to develop the model without a small extension of the road and drains in the soil layer because of the mesh geometry. This extension is surrounded in red in the figure. The extension induced artefact in results, but without any further implications for the upcoming analysis. Therefore we decided not to include these results. In 20% and 30% slope models, the slope is steep enough to develop the model without this extension.

[Figure]

**Again repetitive, not to the point, missing explanations. What is meant with "observed in the same transect". It is unclear to what the sentence in l. 318-319 refers to. Explain what is meant with "the difference along the transect is smaller" (l. 320). The message of the last sentence (l. 322-324) is unclear.**

"observed in the same transect" means observed along the transect formed by the observation section A, B, C, D, E, F, G and H. In other words, it means the simulated flow rates downslope the road in a same model.

"the difference along the transect is smaller" means that difference between G and C observation sections is smaller in a specific model than in another.

For the line 322-324, we wanted to say that the slope increases the differences between maximum and minimum simulated flow rates downslope the road.

This paragraph was reworked because it is not very clear as you mention. (see line 305 to 321)

**The paragraph seems to repeat the arguments just stated in the previous paragraph. Thereby the numbers given are not identical (l. 333 compared to l. 319-320). In l. 333-335, the authors mention the effect of infiltration of low-conductivity soil layers, but it is not clearly displayed. Can infiltration above/through the road structure occur?**

This paragraph was removed because it was as you mention to repetitive.

**Another possible explanation: observed velocities depend on the distance of the observation points from the road structure. For very low hydraulic conductivities the flow dynamics downstream of the road have already formed similar to those upstream of the road. For high conductivities and thus high flow velocities the distance between the road and the observation points is not big enough to establish the previous flow pattern. Therefore the author should investigate additional observation points and provide velocity profiles (in x-direction) for the different road structures.**

We agree that a profile in x direction is useful to have a better understanding concerning the dynamics (see figure 9 line 386)

**The text is again repetitive, e.g. cut out sentence in l.339). The sentence in l. 345- 346) does not make sense. The preferential pathways are not small-scale processes, they are subject to the heterogeneity of hydraulic conductivity. This can be resolved by continuum scale models, but not if assuming a spatially homogeneous conductivity. Furthermore, "the exact hydraulic head in an individual mini-piezometer" is not a process. I cannot agree with the sentence in l. 346-347; simulation results using a spatially homogeneous conductivity are not an average across preferential flow paths.**

This section was reworked to make it less repetitive and sentence l 345-346 will be clarified. We also agree that "hydraulic head" is not a process. "Processes" will be replaced by "observations". Clearly, an average hydraulic conductivity cannot represent the dynamics in individual flow paths but may represent the average dynamics of multiple flow path and less conductive parts. We will reformulate the sentence accordingly.

**Technical corrections:**

**l. 129: subsurface flows perpendicular -> subsurface flow is perpendicular**
Yes, we corrected "subsurface flows perpendicular" by "subsurface is perpendicular". The corrected sentence is: …another important criterion for the selection of the study areas was that subsurface flow is perpendicular to the road (line 113)

**l. 176: The mathematical representation of the nabla-operator is not fully correct. Please put the partial derivatives in brackets to symbolize its vector character.**

These lines were removed.

**l. 176: modify formulation "with the outside of the simulation domain"**

These lines were removed

**l. 306 if the hydraulic conductivity -> if the hydraulic soil conductivity**

Yes you are correct, it is clearer if we add "soil".

**l. 319: correct "from to 0.017"**

Yes it is a mistake. We removed the useless "to".

**l. 367: rephrase to "both sides of the road where hydraulically connected for all investigated road structures"**

Yes, we corrected, the sentence is "The tracer tests showed that both sides of the road where hydraulically connected for all investigated road structures."

**check references (particularly appearance and positions of doi's) as well as ref in l. 411**

We checked the reference (Deroze 1998), the doi is unusual but it is correct.

[revised manuscript text omitted]

**Reviewer 3**

**We thank you for your time and relevant comments and questions. Please find below our answers and the modified manuscript.**

**Your answer sounds quite promising and I am curious about reading the revision. If you would extent your story according to the listed points, I see potential for an improvement of your manuscript.**

Thank you for the comment.

**However, I still not really see the connection of the tracer test and modelling. I agree that a quantitative coupling (e.g. comparison of simulated and observed concentrations) will be very challenging caused by parameter heterogeneities, which are difficult to capture. Also, I can somehow agree to the argument that you want to provide a general modelling framework. However, this leaves me with the question: Why you Discussion paper incorporate the tracer test at all? How does it support your synthetic model? Besides showing natural heterogeneities, you just prove that a L-drain constitutes a preferential flow path. Isn't that a bit too trivial?**

In our experience it never hurts to have field experiments backing up a modelling approach—actually quite the opposite. Even if it might appear trivial at a first glance we believe there is always value in the field data. Apart from this general consideration, we don't think it is a trivial as mentioned by the reviewer. Below some examples (which we will elaborate in the revised manuscript):

- It is also not all clear how important the natural heterogeneities and preferential pathways are in comparison with the drain.
- The price differences of these engineering structures is significant. Given that the models always need to simplify a system it is in our experience unwise to base decision purely on modelling approaches--- the most convincing approach is a combination of both with a demonstration that the planned systems work as planned, and then the models can help to identify how the proposed system will affect flow under different conditions.
- It could also be that the engineering structure is not well implemented or its execution has not been communicated correctly. This is in fact a very common problem. It is not at all trivial to implement these engineering structures in wetland, as the construction machines cannot leave the road,  access is difficult and there are legal considerations ect. With the field tests we show that these structures can be built and functions as planned.

Finally, we want to highlight that this paper is directed not only towards the scientific community, but also stakeholder and the engineering firms who implement these structures. It it therefore particularly important to demonstrate that model can reproduced the general behaviour observed in the field.

**Moreover, regarding the term novelty, we seem to have a slightly different opinion. For me novelty should be more than the application of an existing model to just a new case. Sure, not all HESS papers present an entirely new model or method, but they should present at least a creative solution or new combination of methods. I encourage you to strongly revise your manuscript by adding some new ideas regarding e.g. drying up of fens or gully erosion (could be also something else). Basically, you should dig a bit deeper, but I am optimistic that you are able to do it.**

In term of novelty we added a range of points as suggested by the reviewer. We agree that more results can be extracted from the modelling approach. It is the first time this topic is treated, and we also want to highlight that physically based models such as the ones we use are not that commonly used. Finally, we want to highlight that HESS also encourages the submission of applied research, as highlighted in the description of the journal:

*"HESS encourages and supports fundamental and applied research that advances the understanding of hydrological systems, their role in providing water for ecosystems and society, and the role of the water cycle in the functioning of the Earth system. "*

In addition, section 3 was deeply reworked (see line 295 to 396) and new subsections were added in which we assess the potential risk of gully erosion. To achieve this, the simulated groundwater flow rate are compared with the maximum flux than can flow in the soil calculated with the Darcy law. If the road structure induces a groundwater flow higher than the soil capacity then gully may occur. For example in the surrounded plot in Figure 1 below, you see that L-drain induces a groundwater flow rate higher than the soil capacity and therefore may induce gully erosion. We can also show the consequences of this groundwater concentration by using simulated surface flow illustrated in figure 2. It can be seen that the groundwater flow concentration causes an increase in surface flow and consequently induce gully erosion.

[Figure]

**Figure 1 : Simulated groundwater velocities 2 m downstream each road structures and each parameter combination with a slope 20%.**

[Figure]

**Figure 2 : Simulated surface flow of the KS2-KD2 model and a slope of 20% for each road structure**

Finally, the simulation results are presented at different distance of the road to have a better assessment of the road impact. We are also able to identify areas in which the soil layer is not fully saturated or on the contrary areas in which runoff occurs. See an example in the figure 3.

**Figure 3: Extent of perturbations due to the l-drain road type: Simulated groundwater flow rates at different distances of the road.**

**Minor comments**

**General: Sometimes you are using spaces between numbers and operators and sometimes not. Please, check the guidelines of the journal.**

Spaces between numbers and units were removed as described in the guideline

**Line 59: Capital "V" for Von Sengbusch. It's the start of a new sentence.**

Capital V was corrected.

**Figure 1: The cross-sectional view suggests that the water could easily pass underneath the road. However, in the text you mentioned that the top soil is very thin so that the road blocks the water flow to a large extent (also indicated by the lower figure). Isn't the figure a bit misleading? I would just increase a bit the size of the road and additionally sketch the impermeable bedrock.**

The size of the road in the figure 1 was increased and impermeable bedrock was added. In this way, the reader will directly understand that the groundwater is blocked upstream the road (see line 55)

[Figure]

**Line 126: "similar" or "comparable" instead of "same" would be a more suitable word in this regard.**

"same" was replaced by "similar".

**Line 131: I would add "bed" to "road bed structures"**

"Bed" was added, now the sentence is: To evaluate the hydraulic connection provided by the road bed structures, tracer tests were carried out (see line 114).

**Line 156: I wouldn't use the term "indirectly indicates". I would write something like "clearly shows". At least, I would skip "indirectly".**

The term "indirectly indicates" was removed and replaced by "clearly shows". Now the sentence is: An increase in EC in piezometers located in the downslope area indicates that the injected salt water flowed from the upslope area to the downslope area below the road and clearly shows a hydraulic connection.

**Line 157: Here, it is the other way around. Instead of writing "this indicates that there is no connection", I would be more careful by writing "this indicates a strongly hampered hydraulic connection".**

Yes it is more finely described if we use "strongly hampered" instead of "no connection". We also removed "finally a decrease in EC is not expected". After correction, the whole sentence is: Conversely, if no changes in EC are observed in piezometers, this indicates a strongly hampered hydraulic connection below the road.

**Line 158: I would delete "and finally a decrease in EC is not expected". (It is just too obvious.)**

"and finally a decrease in EC is not expected" is deleted.

**Figure 3: For me, the cross sectional view is a bit superficial, but I guess this is a matter of taste: Still, the spaces before the question marks should be deleted. Moreover, I would just write "Piezometer" instead of "Mini-piezometer".**

The figure 3 was modified according to your comments. (see line 143)

[Figure]

**Line 163: What does "variable saturated" means? Sometimes saturated, sometimes unsaturated or variable hydraulic parameters? This should be explained more specific (I guess it is a terminology from HGS.)**

Variably saturated means that change in saturation of the soil is simulated. However, is not important to mention that here. To be clearer, we changed "variably saturated subsurface water flow" by "subsurface water flow". The corrected sentence is: First, a 3D base case model representing surface and subsurface water flow in a sloping fen was elaborated. (see line 146)

**Line 166: I would replace "produce a sensitivity analysis and explore their sensitivities in" just by "analyse their impact on". Calling it sensitivity analysis is not really wrong, but for my taste not well fitting.**

In our opinion, it is a sensitivity analysis however, we can change. The suggested sentence is:

*For each model, various slopes, organic soil and road drain hydraulic conductivities were implemented to produce a sensitivity analysis and analyse their impact on the sloping fen flow dynamics (see line 148)*

**Section 2.2.1: I would strongly shorten this section, as it is not really a part of your story. If somebody is interested in the mathematics behind your model, he/she would read the original publication of HGS. I would write a couple of lines mentioning the basic assumptions and methods, but no equations. In case you really want to keep them, I have some minor suggestions:**

Indeed, the section may be reduced (reviewer 2 made the same comment). We keep only the main assumptions and method. (see line 154)

**(i) You should give the equations in the same order as referred to in the text, i.e. 1st Richard, 2nd Saint Venant, 3rd Darcy. Or just mention the diffusion a bit later in your text; (ii) Eq 1 and Eq 2 are modified versions of the Richards and Darcy. This should be mentioned. (iii) Line 176: No need to explain "Nabla". It's the common notation; (iv)Line 178: Commonly, "Uppercase Theta" is used for water content and not for porosity;(v) Line 180: I would add "saturated". K is the "saturated" hydraulic conductivity:(Multiplying with kr results in the actual hydraulic conductivity.)**

These lines were removed.

**Line 207f: "was used on the right face" – left and right are just a matter orientation. Maybe you better write something like: The lowest cells of the slope constitute a constant head boundary condition.**

Yes, it is better to use "the lowest cells of the slope" than "one the right face". The sentence now reads: A constant groundwater head boundary condition (Dirichlet type) equal to the ground surface elevation (2m) was used on the lowest cells of the slope (x=76m on the **Erreur ! Source du renvoi introuvable.**a) allowing the groundwater to flow out of the model. (see line 176)**.**

**Line 218: Missing space between "2" and "m".**

According to the guideline, we should not use a space between a number and an abbreviation of a unit. Therefore, we removed all spaces in the manuscript.

**Line 234: Generally, I prefer the use of SI units, i.e. m/s instead of m/d.**

As hydrogeologist, we also prefer m/s instead of m/d, however, the manuscript is not only hydrogeologist but for other environmental sciences such as biology. In our opinion, m/d provides greater clarity.

**Line 256f: What do you mean by "length scale of one to several meters". Is this a common expression?**

"length scale of one to several meters" is not a common expression but a mistake. We removed "length scale". Now the sentence is: In contrast, the EC maps established prior to the tracer test show a spatial variability of one to several meters (line 238)

**Line 257: "629uS/am" – What is this? I guess 629 _S/cm**

Yes it is 629µS/am. We corrected. (line 240)

**Line 279: "local drying up of the soil)" – If you consider this as a problem, it would be quite easy to further investigate it with your numerical model. This would allow answering the question: how large is the affected area and to which extent it dries out?**

Yes, you are absolutely right. Therefore, a new figure (figure 10 line 365) was created in which we can see the extent of perturbations induced be the l-drain structure.

**Figure 7: In column 2 and 3 you are showing EC values. I am wondering why you are using totally different graphical representations. Moreover, if you are interpolating (I am not a big fan of interpolation, if it is not really necessary: : :), you should state which method you are using. What kind of background map you are using? Does it tell us something?**

The spline interpolation was used for the 2$^{nd}$ column (we added this information in the legend of the figure). We used a different representation bcause it is not very clear if we use the interpolation in the 3$^{rd}$ column (see figure below). The backgroung does not tell something.

[Figure]

**Line 288-292: For me, these lines are superficial. I would just delete them.**

We wanted to help the reader by describing each step of the result interpretation. If you think it is superficial, we can remove them.

**Line 293-301: This is very trivial and doesn't need any explanation. It can be directly derived from the Darcy equation (at least for the base case model).**

Yes it is trivial it can be directly derived from the Darcy equation. However, it seems important to describe the base case insofar as the base case is used to compare other results.

**Line 316f: Are you sure that "may be" is the right expression here?**

Do you prefer "can be"? We modified the "may be" by can be".

**Figure 8-10: It is not very comfortable to analyse the differences between the different slopes. Can't you just put all figures together using a slope specific colour?**

Yes it is true. We grouped together the three slopes.

**Line 451: Is the year 2005 correct? I guess you want to refer to the manual, or? The one, I found, is from 2010.**

[revised manuscript text omitted]

---

## Referee Report (RR1)

Second Revision on „Assessing the perturbations of the hydrogeological regime in sloping fens through roads" by Fabien Cochand, Daniel Käser, Philippe Grosvernier, Daniel Hunkeler, Philip Brunner

**General Comments**

I appreciate the detailed response of the authors to the comments I raised in the first round of review. They addressed all points and adapted the manuscript accordingly in most places. The manuscript and figures have been improved significantly.

At some positions, the added text requires further revision. Sometimes, the authors gave explanations as response to the reviewer which should be given in the manuscript to clarify these points also for the reader who might wonder at the same aspects while reading. There are a also few remaining open questions from the first round of review. These issues are addressed in the comments below.

Although the author stated that they reworked the text (specifically in some sub-sections of 2 & 3), it appears to me that they only added few lines/words/brackets at critical points for some parts. Several paragraphs are still written in a repetitive and elongated manner which is not reader friendly. You could easily cut out redundant phrases and repetitions to increase the readability (some examples given below). The authors should consider professional language support or at least a proper proofreading and revision by a native speaker.

**Specific Comments**

- Background information on the three road structures developed in Switzerland is still missing [introduction].
- Typo in l. 71-72
- Integrate your response to the text as background information on the model setup for the road types, e.g. When a road construction takes place, impermeable material is excavated upstream and filled downstream which is represented by an increased number of inactive cells below the road.
(from answer to "The mesh modifications for cases 5d, 5e and 5f show an artificial increase of inactive cells below the road (step shape instead of continuous slope form). Shouldn't there be soil cells below the road construction? This might significantly modify the simulation results.") [section 2.2.2]

Section 2.2.3:
- it is not done by just renaming the subsection title; the text should be adapted as well (e.g. the first sentence in the section still starts with "The sensitivity analysis..." )
- many newly added sentences require improvement in language (line 212-216, l. 2018-220), please perform a proper proofreading
- The authors still have a lot of redundant text which inhibits the readability:
  - Phrases like "In order to simulate each parameter combination" (l.224) could easily be cut out without any loss of information.
  - You could cut the entire sentences in l. 229-232: the method section should contain the specific information, not elaborate explanations on the motivation (which is anyway clear at that point of the paper).

Section 3
- Figure 7:
  - First column: Head profile for the second and third site are still missing head values above and below the road which inhibits a proper picture of the hydraulics at these sites.
  - I agree with the authors that the original form of display is preferable.
- Section 3.2 could still be condensed to focus on key facts and major results.
- The discussion on gully erosion is a valuable addition. However, the text requires proper proofreading and shortening (e.g. the sentence in l. 341 is basically redundant).
- Typo in l. 322 "and"
- Figure 11: is interesting, however only for the no-road and L-drain comparison. The authors might consider different scaling to see differences also in the no-excavation and wood-log structures.
- Specific recommendations (l.335 ff) are rather part of the conclusion section.
- L 344-358 are also rather part of a summary and/or conclusion.
- I still cannot agree with the sentence "Models results have to be interpreted as an average across multiple preferential flow paths." (l. 353-354) The simulation results in a homogeneous medium do not represent mean results of simulations in heterogeneous domains with preferential flow path! (Maybe just skip the sentence, the previous one give a proper explanation.)

---

## Author Response (AR2)

Dear Professor Hildebrant,

Following our mail exchange and the clarifications during skype conversation, we have modified the manuscript in the following way:

- Consideration of the comments of reviewer 2 (see below for detailed answers)
- Explicit discussion that the modelling allows to assess a "worst-case scenario" and a relative ranking of the potential impact
- Significant tightening of the text in response to the comments of reviewer 2

After careful consideration we decided to keep the current order of discussing the field experiment first and then the modelling.

Thank you again for handling the paper.

**Answers to your specific comments in your mail from the 23.8.2019:**

**For example, in section 3.1 the manuscript states that one of the preferential pathways at site SCH was due to the L-drain and the other was unrelated. This may suggest that the L-drain creates flow paths that are of similar impact as the natural heterogeneity. Or worse, both flow path could actually be due to natural heterogeneity.**

The natural heterogeneity under the road no longer exists. The road was constructed in a way to concentrate flow through the drain. This is exactly what we see in the example of SCH.

**In my reading of section 3.1 and presented data, the field study does not contribute to "show" that the L-drain constitutes the largest perturbation (this is a statement from the abstract), and also not that modeling and field study are coherent (from the abstract). I agree that the field observations do not contradict the modeling study, but this is a substantially more careful phrasing as currently.**

Following your suggestion, we have adjusted the legend in Figure 7 to highlight the relative importance of the drains. The field results clearly show that the relative impact (in terms of concentration the flow) is most pronounced with the L-drain. We have added explanations highlighting that the models present a "worst-case" scenario. This is also considered in the abstract and in the conclusions.

**The main conclusion of the abstract is that L-drains constitute the largest perturbation to the ground water flow, and the other investigated structures less so. My main criticism is that the word "perturbation" implies moving the system away from its natural state. The natural state is one marked with substantial heterogeneity causing flow paths. That natural heterogeneity is substantial, as shown be the experiment.**

As discussed with you on skype, we feel the word perturbation is appropriate as a constructed road through a wetland is always moving the system away from its natural state.

**In my letter, I also proposed a way forward which does not imply new model runs. In response in your e-mail below you state "heterogeneity can, if one is lucky, reduce the influence of the drain". This type of statement should absolutely enter the manuscript. The homogeneous case is more or less the worst-case scenario. This is ok. But the overall tone of the manuscript needs to reflect this. Also, how can you quantify "if one is lucky", based on the field study?**

We discuss this in the context of Figure 7. For the L-drain case one can see that a "plume" is forming downstream of the drain, i.e. that the concentrated flow is, to a certain extent redistributed. If no gullies form, it is indeed possible that the influence of the road downstream is reduced due to the horizontal redistribution of water through heterogeneous pathways.

**Personally, I would switch the order of presentation to first show the model results and follow up with the field study and discuss how it actually supports the model conclusions and where it is maybe inconclusive.**

We have carefully considered this point and decided to stick to the original order. However, we added an additional explanation concerning the field- and modelling approach which also better explains the order.

**Reviewer 2**

**Second Revision on „Assessing the perturbations of the hydrogeological regime in sloping fens through roads" by Fabien Cochand, Daniel Käser, Philippe Grosvernier, Daniel Hunkeler, Philip Brunner**

**General Comments**

**I appreciate the detailed response of the authors to the comments I raised in the first round of review. They addressed all points and adapted the manuscript accordingly in most places.**
We thank the reviewer for this positive feedback.

**The manuscript and figures have been improved significantly. At some positions, the added text requires further revision. Sometimes, the authors gave explanations as response to the reviewer which should be given in the manuscript to clarify these points also for the reader who might wonder at the same aspects while reading. There are also few remaining open questions from the first round of review. These issues are addressed in the comments below.**
See our response to the comments below. We have gone the text very carefully and tightened the text and presentation in several places.

**Although the author stated that they reworked the text (specifically in some sub-sections of 2 & 3), it appears to me that they only added few lines/words/brackets at critical points for some parts. Several paragraphs are still written in a repetitive and elongated manner which is not reader-friendly. You could easily cut out redundant phrases and repetitions to increase the readability (some examples given below). The authors should consider professional language support or at least a proper proofreading and revision by a native speaker.**
The document has been carefully checked and the wording has been improved.

**Specific Comments**

**Background information on the three road structures developed in Switzerland is still missing [introduction].**
We are aware of this. However, such data are not available. Most road constructions are on private grounds and there is no central data-based bringing together all of these data.

**Typo in l. 71-72**
Thank you

**Integrate your response to the text as background information on the model setup for the road types, e.g. When a road construction takes place, impermeable material is excavated upstream and filled downstream which is represented by an increased number of inactive cells below the road. (from answer to "The mesh modifications for cases 5d, 5e and 5f show an artificial increase of inactive cells below the road (step shape instead of continuous slope form).**
We have now included a slightly modified version of this sentence in the introduction and the model setup:

**Shouldn't there be soil cells below the road construction? This might significantly modify the simulation results.")**
No, in the cases we know this is not the case. The roads are constructed in a way to avoid this. The depth of the road construction does not have to go deep, as the soil layer above the clay is very thin (e.g. 40cm) in the sloping fens in Switzerland.

**Section 2.2.3:**
**It is not done by just renaming the subsection title; the text should be adapted as well (e.g. the first sentence in the section still starts with "The sensitivity analysis...")**
As highlighted above, we have modified and tightened the text.

**Many newly added sentences require improvement in language (line 212-216, l. 2018-220), please perform a proper proofreading**
As highlighted above, we have modified and tightened the text.

**The authors still have a lot of redundant text which inhibits the readability:**
As highlighted above, we have modified and tightened the text.

**Phrases like "In order to simulate each parameter combination" (l.224) could easily be cut out without any loss of information.**
**You could cut the entire sentences in l. 229-232: the method section should contain the specific information, not elaborate explanations on the motivation (which is anyway clear at that point of the paper).**
As highlighted above, we have modified and tightened the text.

**Section 3**
**Figure 7: First column: Head profile for the second and third site are still missing head values above and below the road which inhibits a proper picture of the hydraulics at these sites. I agree with the authors that the original form of display is preferable.**
Note that the hydraulic head downslope the road in the Stouffe site is about 25cm and upslope the road in the Schöniseischwand is about 225cm and. The isolines are drown each 50cm, therefore these isolines are not presented in the figure.

**Section 3.2 could still be condensed to focus on key facts and major results.**
We have condensed this section

**The discussion on gully erosion is a valuable addition. However, the text requires proper proofreading and shortening (e.g. the sentence in l. 341 is basically redundant).**
The section was carefully reworked and shortened

**Typo in l. 322 "and"**
Thank you

**Figure 11: is interesting, however only for the no-road and L-drain comparison. The authors might consider different scaling to see differences also in the no-excavation and wood-log structures.**
We have decided to remove Figure 11 and added a brief description to the text. Even with additional scaling, the perturbations are minor and thus can be discussed in the text.

**Specific recommendations (l.335) are rather part of the conclusion section.**
We have removed this information from this section

**L 344-358 are also rather part of a summary and/or conclusion.**
Parts of this section were moved to the conclusions. We kept the model-specific point only

**I still cannot agree with the sentence "Models results have to be interpreted as an average across multiple preferential flow paths." (l. 353-354) The simulation results in a homogeneous medium do not represent mean results of simulations in heterogeneous domains with preferential flow path! (Maybe just skip the sentence, the previous one gives a proper explanation.)**

Thank you, we have deleted this sentence, as you suggested.

[revised manuscript text omitted]
). ToTo construct the foundation of the a road, amaterial with a very low permeability is used.used material under the road. This subsequently blocks the flow from the upslopedownstream. However, due to the buildup of hydraulic heads in the upslopewithout the presence of a drain to connected the upstream and the downstream,is can be innundated during precipitation events.  To reduce the occurrence of inundations, drains are installed under all roads (Figure 1c). The design and the materials of drains havehave potentially a significant effecteffect on flow dynamics. Figure 1c presents a typical condition where a non-continuous drain (i.e., drains are perpendicularly installed at regular distances along the road) is installed.installed. The drain captures the  flow upslope along the road and the discharge is released in a concentrated manner downslope.

This  The concentration of  flow downslope  may induce gully erosion and disturb the hydraulic regime of the sloping fens. For example, the wetland is at risk of drying out downslope of the road as the flow is concentrated to a small strip downslope of the drain. Note however, that a gully must not necessarily develop because the flow-velocity at the drain-exit might not be sufficiently large to trigger erosion. Also, the drying out of the wetland beyond the direct vicinity of the downslope area of the drain must not necessarily happen. The The concentrated release from the drain can water, to a certain extent, spread out horizontally. In any case, as

Aa road  constitutes a hydrogeological barrier which which perturbs the natural  flow dynamics.

[revised manuscript text omitted]

ATo numerically solve the 3D flow equation, aA 3D- finite element mesh was developed (Figure 5Figure 5a). The mesh is 76m long in the X direction, 20m in the Y direction and the mesh thickness is 1.2m. The top elevation was fixed at 2m on the right side (x=76m) and varies from 9.6m to 24.8m on the left side (x=0) according to the slope of the model. The mesh was composedmade upcomposed of 24 layers, 127,200 nodes and 118,440 rectangular prism elements. To guarantee numericalensure an appropriate level of detailnumerical stability, several mesh discretization refinements were implemented. Themadeimlemented. Therefore, tThe element size varies between 2m and 0.1m horizontally (in the X and Y directions) and 0.09m and 0.06m vertically.

The base case model and the three other models representing different road types have the same boundary conditions and finite element meshes, however, modifications were made between coordinates 61<x<66 to for the implementation of the different road types. Figure 5Figure 5 depicts the differences between the base case model (Figure 5Figure 5a and b) and models with roads (Figure 5Figure 5c, d, e and f). In the case of models with simulating a road, the mesh and the material properties waswere deformed ajustedand the properties were adjustedchanged. The fine spatial discretization of the mesh created between the coordinates 61<x<66 allows a more accurate representation of the simulated processes where high hydraulic gradients are expected (near roads and drains). Additionally, the refinements allow an accurate representation of drains and the roads.

[Figure]

**Figure 5 : Model development: a) Base case model, b) Base case model cross-section between 61m < x < 66m, c) No-**
**excavation model between 61m < x < 66m, d) L-drain model between 61m < x < 66m, e) L-drain model between 61m <**
**x < 66m along the transversal drain f) Wood-log model between 61m < x < 66m.**

**2.2.3**   **Model application**

The model applicationsensitivity analysis consists of the variation of model properties. and parameters in order to assess their effect onunderstand how they control the groundwatersloping fen dynamics. The sensitivities of the following parameters were analyzed: fen slope, soil hydraulic conductivities and road drain hydraulic conductivities. These parameters were selected because according to thethey govern thethe Darcy's law (1)

theyand consequentlythey control the groundwater flow dynamics. K is the hydraulic conductivity of the soil and the drain and ∇H the hydraulic gradient ofgradient oof the fens which itselfitslef iswill be strongly influenced controlled by the topographical slope.

$$q = K * \nabla H \qquad (1)$$

For each property varied in the sensitivity analysis, three different values were chosen (Table 2), ): a low, an-intermediate and a high value.values with the aim of covering the whole range of its observed values in sloping fens. For the soil hydraulic conductivities (KS), values presented in Charman (2002)Charman (2002) were used and vary varied between 8.64m/d and 0.0864m/d. This corresponds to a soil composed of gravely, organic matter (as observed for example in St-Antonien site) or loamy organic matter (as observed for example in

Schoeniseischwand site). α and β Van Genuchten parameters (α and β), )and the residual water content, as well as the residual water content, were considered similar assumingwere not varied. their capillary rises are comparable and does not play a critical role in a 40cm soil layer mainly saturated. The road drains (KD) which are made with of coarse or very coarse gravel wereand havewere assigned a hydraulic conductivity varying between 8640m/d and 86.4m/d (Fetter 2001).) and, their van Genuchten parameters are corresponding to those of gravel. The slopes were fixed at 10%, 20% and 30%, as observed during the fieldwork. TheNote that tThe hydraulic conductivities of the wood-log (W-L) drain hydraulic conductivities of the wood log (W-L) were assumed ten times more conductive and more porous than the gravel drain. because of its particular structure (wood logs). The road concrete is almost impermeable and was thus conceptualized with a very low hydraulic conductivity. and its van Genuchten parameters corresponding to ofa fine material. The road basement is constructedmade withconstructed using highly compacted fine material (sand and loam) andhave a lowfeatureand was thus implemented with a low hydraulic conductivity, the and are assigned van Genuchten parameters of corresponding to fine material. Finally, the implemented soil and road surface flow properties correspond to a wetland and urban cover (Li et al., 2008)(Li et al., 2008).

*Table 2 : Subsurface and surface flow parameters.*

| Subsurface flow properties | | | | | |
|---|---|---|---|---|---|
| | Hydraulic conductivity | Porosity | Van Genuchten α | Van Genuchten β | Residual water content |
| Units | K [md⁻¹] | θ [-] | α [m⁻¹] | β [-] | Swr [-] |
| Soil - KS1 | 8.64 | 0.25 | 4 | 1.41 | 0.04 |
| Soil - KS2 | 0.864 | 0.25 | 4 | 1.41 | 0.04 |
| Soil - KS3 | 0.0864 | 0.25 | 4 | 1.41 | 0.04 |

| | | | | | |
|---|---|---|---|---|---|
| **Drains - KD1** | 8640 | 0.25 | 29.4 | 3.281 | 0.04 |
| **Drains - KD2** | 864 | 0.25 | 29.4 | 3.281 | 0.04 |
| **Drains - KD3** | 86.4 | 0.25 | 29.4 | 3.281 | 0.04 |
| **Drains - WL - KD1** | 86400 | 0.7 | 29.4 | 3.281 | 0.04 |
| **Drains - WL - KD2** | 8640 | 0.7 | 29.4 | 3.281 | 0.04 |
| **Drains - WL - KD3** | 864 | 0.7 | 29.4 | 3.281 | 0.04 |
| **Road concrete** | 0.0000864 | 0.05 | 1.581 | 1.416 | 0.04 |
| **Road basement** | 0.00864 | 0.25 | 4 | 1.416 | 0.04 |

| **Surface flow properties** | | | | |
|---|---|---|---|---|
| | **Coupling length** | **Manning's roughness coefficient** | | **Rill storage height** | **Obstruction height** |
| **Units** | $l_c$ [m] | $n_x$ [$m^{-1/3}s$] | $n_y$ [$m^{-1/3}s$] | $D_t$ [m] | $O_t$ [m] |
| **Soil** | 1. x $10^{-2}$ | 0.03 | 0.03 | 0.005 | 0.005 |
| **Road** | 1. x $10^{-2}$ | 0.018 | 0.018 | 0.001 | 0.001 |

In order to simulate each parameter combination, a total of 90 models were developed (27 models for each road structures and 9 models for natural conditions). Models are run for 10'000 days (about 27 years) with a constant flux equal to 380mm/y on the top representing the rainfall to reach a steady state.

Subsequently, subsurface flow rates in the soil layer were extracted at each section with an area of 0.4m² (1m wide times the soil thickness) presented in Figure 6.

Changes in subsurface flow rates indicate a perturbation of flow dynamics and therefore, a comparison of flow rates between each model was made to present the effect of each road structure and sloping fen properties on the dynamics.

[Figure]

**Figure 6 : Location of observation  sections in the models.**

**3     Results and Discussion**

**3.1     Fieldwork**

Based on the observations, all sites show a continuous saturated zone before the experiment, both upslope and downslope of the road, the hydraulic gradients being  similar to the terrain slope (Figure 7, 1st column). In contrast, the EC maps established prior to the tracer test show a spatial variability of one to several meters (Figure 7, 2nd column.). Within each plot, EC varies from 482 to 629µS/cm. At the SCH site, the highest values are located downslope of the L-drain outlet which could indicate that the EC increases as water is flowing through the drain (e.g. through the dissolution of the construction material). Given that this initial distribution of EC is not uniform, the comparison of EC after the sprinkling experiment has to be made in a relative manner (Figure 7, 3rd column).

The heterogeneity of the hydraulic conductivity of the soil is apparent from the tracer tests results (Figure

7Figure 7, 3rd column: EC 24 hours after injection). At all four sites, the front of the saline solution is not uniform because of the but follows the heterogeneity of the soil hydraulic conductivity. Nevertheless, theroad structuresthe road structures or the drains may createplay the role of aconstitutecreate preferential flow paths. This thatThis is clearly occurringparticularly obviousclearly occuring 
[revised manuscript text omitted]

In addition, the perturbation on the roads upslopeuphill of the road was assessed. Although the formation of gullies depends of a lot of other factors (Valentin et al., 2005b), such as soil type or the rain intensity, the model showed that downstream L-drain structure may cause runoff concentration which is an important factor.

A simple recommendation can be made to avoid this runoff concentration.

If the maximum flow rate capacity of the soil is smaller than the flow rate induced by precipitation, the installation of an L-drain structure should not be considered.may lead to surface runoff.

If the maximum flow rate capacity of the soil is larger than the flow rate induced by precipitation, an L- drain may be considered only if the concentred flow calculated by multiplying the drainage area by the precipitation is smaller than maximum flow rate capacity of the soil

Finally, the impact of road structure on the upslopeupstream road dynamics wasmaywere be also assessed (Figure not shown)). Figure 11Figure 11 shows the same information as Figure 8Figure 8 but at 2.5m upslope.

Upslopeupstreamhill. It can be seen that for all models, uUpstreamhill flows are similar to the base case model, thus the influence of the road is, not unexpectedly, marginal for all road types. . This means that all structures allow the groundwater to crossflow across the road.

The

The significant impact of the L-drain road structure which concentrates groundwater flow is clearly establishedidentified in the numerical approach and is consistent with the field observations. For the other road structures alsotoo, the numerical models are consistent with fieldwork results by showing a relatively undisturbed groundwater flow downslope the road. The use of numerical models allowed for a quantitative estimation of the flow perturbation induced by each road structure and model results were consistent with the field observations. In addition, tThe development of models with various combinations of parameters also allowed for exploring a larger parameter space than using field work only. For instance, the fact that the impact of an L-drain structure on the water dynamics is less marked if the hydraulic conductivity of soil is low would have been impossible to identify by using fieldwork only. However

[revised manuscript text omitted]